# Lin28, a major translation reprogramming factor, gains access to YB-1-packaged mRNA through its cold-shock domain

Anastasiia Samsonova[1], Krystel El Hage [1], Bénédicte Desforges[1], Vandana Joshi[1], Marie-Jeanne Clément[1], Guillaume Lambert[1], Hélène Henrie[2], Nicolas Babault[2], Pierrick Craveur[2], Rachid C. Maroun[1], Emilie Steiner[1], Ahmed Bouhss [1], Alexandre Maucuer [1], Dmitry N. Lyabin[3], Lev P. Ovchinnikov[3], Loic Hamon [1] & David Pastré [1✉]

The RNA-binding protein Lin28 (Lin28a) is an important pluripotency factor that reprograms translation and promotes cancer progression. Although Lin28 blocks let-7 microRNA maturation, Lin28 also binds to a large set of cytoplasmic mRNAs directly. However, how Lin28 regulates the processing of many mRNAs to reprogram global translation remains unknown. We show here, using a structural and cellular approach, a mixing of Lin28 with YB-1 (YBX1) in the presence of mRNA owing to their cold-shock domain, a conserved β-barrel structure that binds to ssRNA cooperatively. In contrast, the other RNA binding-proteins without cold-shock domains tested, HuR, G3BP-1, FUS and LARP-6, did not mix with YB-1. Given that YB-1 is the core component of dormant mRNPs, a model in which Lin28 gains access to mRNPs through its co-association with YB-1 to mRNA may provide a means for Lin28 to reprogram translation. We anticipate that the translational plasticity provided by mRNPs may contribute to Lin28 functions in development and adaptation of cancer cells to an adverse environment.

[1] SABNP, Univ Evry, INSERM U1204, Université Paris-Saclay, 91025 Evry, France. [2] SYNSIGHT, 4 rue Pierre Fontaine, 91058 Evry, France. [3] Institute of Protein Research, Russian Academy of Sciences, Pushchino 142290, Russian Federation. ✉email: david.pastre@univ-evry.fr

After transcription, splicing, and nuclear export, mature messenger RNAs (mRNAs) are ready for translation, but not all of them are translated after they enter into the cytoplasm[1,2]. Some mRNAs are indeed stored in the cytoplasm while waiting for the proper time and location for their activation. The spatio-temporal control of mRNA translation is required to enable complex cellular processes such as those that occur during embryogenesis and axon genesis[3]. In addition, translation regulation enables a rapid response to various stimuli by activating specific mRNAs[4] without requiring de novo transcription. To keep mRNAs in a dormant state, mRNAs are packaged into ribonucleoprotein complexes making them inaccessible to ribosomes[5] (referred to herein as "mRNPs"). A major protein component of mRNPs is YB-1 (Y-box-binding protein, YBX1 gene), an abundant mRNA-binding protein in the cytoplasm[1]. YB-1 has the ability to polymerize nonspecifically along mRNA to form untranslatable beads-on-a-string structures[6] but also to unfold mRNAs into translatable nucleoprotein filaments[7], when activation takes place, possibly after YB-1 phosphorylation[5]. However, little is known about other mRNA-binding proteins (RBPs) that should interact with mRNPs to regulate specifically their repression/activation and routing.

Here we explore the structure-function relationship of mRNPs associated with Lin28. Lin28 (Lin28a) and also its paralogue Lin28b are important reprogramming factors expressed during embryonic development and are associated to pluripotency[8]. In addition, while Lin28 is generally not present in mature tissues, Lin28 is re-expressed in several cancers to support cancer cell growth[9–11] and resistance to cancer therapies[12,13]. To explain Lin28 functions in stem and cancer cells, many studies have focused on let-7, a microRNA controlling the expression of genes related to differentiation and growth[14]. Lin28 notably inhibits the processing of pri-let-7 thereby preventing differentiation[15]. However, several lines of evidence indicate additional roles for Lin28 besides its association with the let-7 pathway[16–18]. For instance, during neurogliogenesis in vitro, Lin28 expression occurs prior to any inhibition of let-7 expression and blocks glycogenesis independently of let-7 accumulation[19]. More importantly, endogenous Lin28 binds to thousands of mRNAs, whether in stem[20–22] or cancer cells[23], whereas the binding of Lin28 to let-7 represents only a small fraction of the RNA:Lin28 complexes. Albeit the Lin28/let-7 axis is surely important in translation regulation, a more global role of Lin28 in controlling the translation of many mRNAs is very likely. Some let-7-independent functions of Lin28 have already been proposed in the processing of transcripts regulating glucose metabolism[8] or membrane functions[22] that are associated to pluripotency and cancer growth.

However, the molecular mechanism by which Lin28 globally reprograms cell translation independently of let-7 remains puzzling. One of the scenarii which constitute our working hypothesis is the preferred association of Lin28 to mRNPs. A mechanism that directs Lin28 to YB-1-packaged mRNPs would enable Lin28 to turn on/off the translation of many mRNPs. Targeting mRNPs constitutes an efficient and easier mean to control translation, rather than acting on the preinitiation step of translation or the rate of protein synthesis in polysomes. In support for this hypothesis, Lin28 may interact with YB-1 through their common cold shock domain (CSD) by binding cooperatively to RNA and single-stranded (ss) DNA[7,24]. The CSD originates from cold-shock proteins in bacteria. This highly conserved domain allows bacterial resistance to low temperatures due to its capacity to multimerize along mRNA. Otherwise, mRNA secondary structures would block translation at low temperatures[25]. Interestingly, the ability of CSD to multimerize along mRNA is also preserved in YB-1 CSD[26]. In addition, both Lin28 and YB-1 have a positively charged C-terminal domain (CTD) following their CSD

(8 and 7 positive charges for Lin28 (aa 122–135) and YB-1 (aa 137–152), respectively). Flanking the CSD, this unstructured domain can bridge consecutive CSDs along the mRNA, as observed in the linear mRNA nucleoprotein filament formed by YB-1 in vitro[7]. Using microtubules as intracellular nanoplatforms to probe the co-localization between RNA-binding proteins[27] and their mixing[28] (microtubule bench assay), we showed that Lin28 and YB-1 co-localize in cells thanks to their common CSD, unlike the other tested RNA-binding proteins, G3BP1, FUS, TDP-43, LARP6 and HuR that do not have CSD (Supplementary Fig. S1b). NMR spectroscopy further revealed the molecular mechanism responsible for the cooperative association of Lin28 and YB-1 to single-stranded nucleic acids. The intramolecular interaction of residues located in the CTD and CSD loop 3 in Lin28 or YB-1 enables the deployment of the positively charged CTD. The CTD is oriented towards the negatively charged sugar-phosphate backbone of nucleic acid strands that interact with an adjacent CSD, thus bridging consecutive CSDs. Using Lin28 mutants to alter the mixing between Lin28 and YB-1 in mRNPs, we explored the relevance of the co-association of YB-1 with Lin28 in mRNPs in a cellular context. We notably found a cooperative association of Lin28 with YB-1 in stress granules, that are liquid-phase mRNA compartments[29] formed after translational arrest, and a YB-1-dependent control of cell proliferation exerted by Lin28 in HeLa cells. The analysis of gene expression across tissues and during embryonic development also point towards a functional link between Lin28 and YB-1 in vivo. In light of these results, we propose a mechanistic model for the interaction between Lin28 and YB-1 that should be useful for further exploring the let-7-independent contribution of Lin28 to cancer[30] and neurological diseases[31].

## Results

**Lin28 co-localizes with YB-1 and mixes with YB-1-rich compartments but not with other RNA-binding proteins, FUS, HuR, G3BP-1, LARP-6, in HeLa cells.** To probe the co-localization between Lin28 and YB-1 in the cytoplasm, we used the microtubule network as an intracellular bench[27]. Briefly, a bait protein is brought on microtubules following the expression of a fusion protein comprising a RFP-labeled protein, a linker and a microtubule-binding domain (MBD). Cells also co-express a GFP-labeled prey protein, the presence of which on the microtubules reveals a co-localization with bait proteins (Fig. 1a). This method gave us the first hint of a putative direct or indirect interaction between Lin28 and YB-1, which was also supported by co-immunoprecipitation assays[27]. To confirm these results and understand the nature of this interaction, we extended this study by combining 4 RBPs used as baits (Lin28, YB-1, and two proteins without CSD, FUS and G3BP-1) and 4 RBPs used as preys (G3BP-1, Lin28, YB-1, CSDE1, Fig. 1b and Supplementary Fig. S1c). Lin28, YB-1, CSDE1 have at least one CSD (CSDE1 has 5 CSDs[32]). In the absence of the bait protein, the prey proteins are homogenously distributed in the cytoplasm and are not present on microtubules, which is a prerequisite for using this method (Supplementary Fig. S2d). The co-localization score measured on microtubules shows that proteins with CSDs co-localize with each other (see red dots, Fig. 1b), even if to a lesser extent for CSDE1, whereas CSD proteins poorly co-localize with FUS and G3BP-1 that have no CSD.

We then questioned whether Lin28, through its ability to co-localize with YB-1, could mix with YB-1-rich compartments. To this end, two RBPs are fused to a microtubule-binding domain to generate mRNA-rich compartments along microtubules as previously reported for TDP-43, FUS, HuR and G3BP-1[28]. Compartmentalization of RBPs can be due to homotypic interactions that

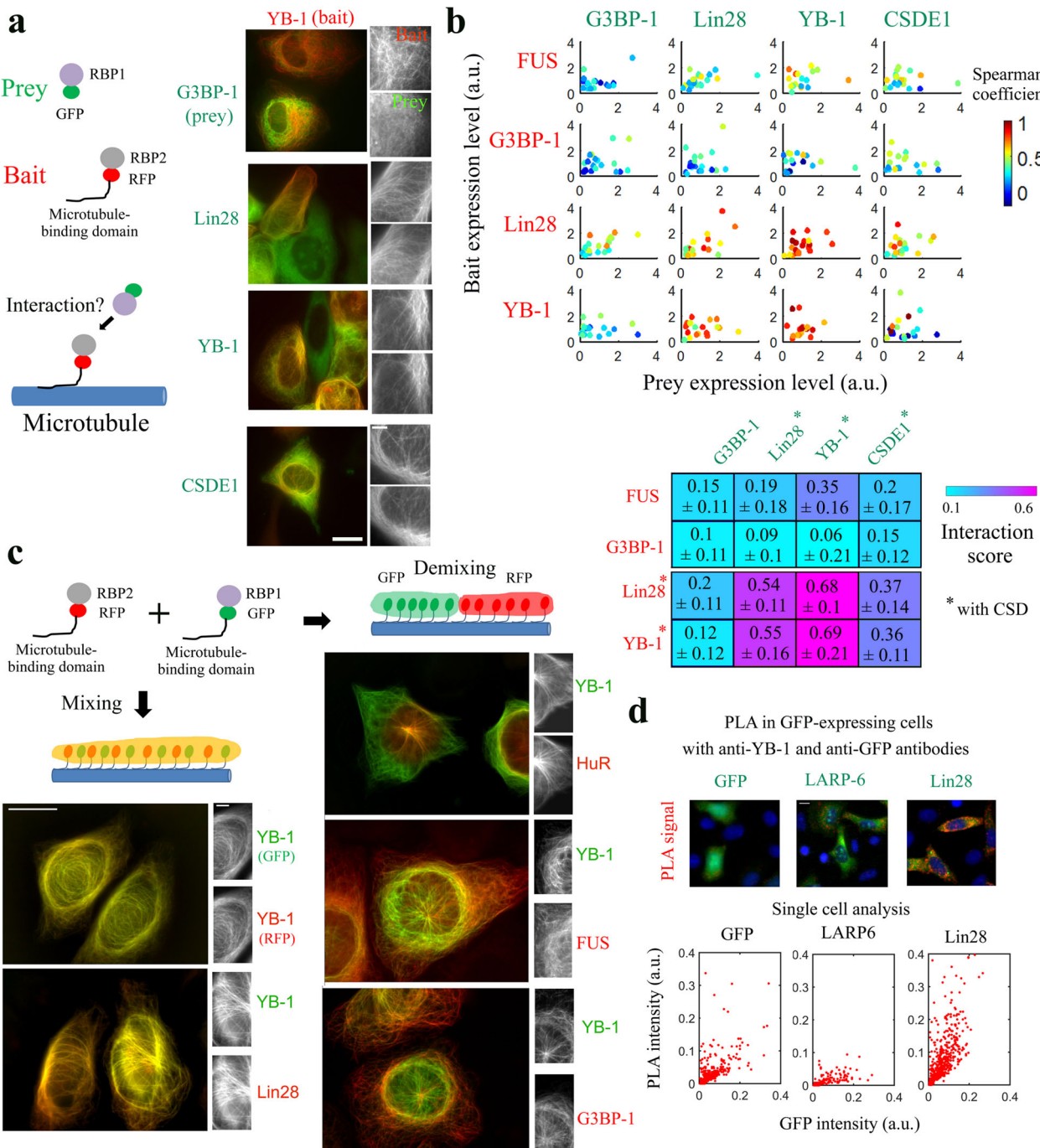

**Fig. 1 Lin28 and YB-1 colocalize in cells and mix along microtubules, in contrast to G3BP-1, HuR, or FUS. a** Left panel: Detection scheme of the microtubule bench assay. Right panel: YB-1 used as bait (in red) and 4 RNA-binding proteins (RBPs) used as preys (in green) were co-expressed in HeLa cells. The bait is brought onto microtubules owing to its fusion to a microtubule-binding domain (MBD). The presence of a prey on microtubules therefore reveals a bait/prey interaction. Scale bar: 15 μm and 4 μm (higher magnification, right panel). **b** Upper panel: Spearman's coefficient reflecting the presence of the prey on microtubules was measured at the single cell level ($n = 20$) for 4 different baits and 4 different preys, as indicated. Lower panel: Interaction score for indicated preys and baits measured by extrapolating the Spearman's coefficient for very low bait expression level (see Materials and Methods). Values are given with 95 % confidence bounds. **c** Two RBPs as indicated are confined on the microtubule network (fused to RFP/GFP-MBD) to visualize their mixing/demixing in HeLa cells. Mixing: yellow microtubules. Demixing: red and green microtubules. Scale bars: 15 μm and 4 μm (higher magnification, left panel). **d** Upper panel: Representative images for Proximity Ligation Assays (PLA) between GFP and endogenous YB-1 in HeLa cells expressing indicated proteins GFP, LARP6-GFP and Lin28-GFP. Lower panel: PLA signal versus the expression levels of indicated proteins at the single cell level (GFP integrated intensity, $n > 100$). Scale bar: 15 μm.

may notably take place in low complexity domains (TDP-43, FUS, and possibly the C-terminus of YB-1), mRNA base-pairing and mRNA bridging by multiple RNA-binding domains (HuR, Lin28; Supplementary Fig. S1b)[28]. We first controlled that both YB-1 and Lin28 can be brought on microtubules to form mRNA-rich compartments (Supplementary Fig. S2a and[27]). Then, we observed that YB-1 does not mix with FUS, G3BP-1 or HuR, revealing thereby that mRNA-rich YB-1 compartments tends to be separated from three RBPS without CSD (red or green microtubules, Fig. 1c). However, strikingly, Lin28 and YB-1 are mixing pretty well along the microtubule network (yellow microtubules).

To confirm the co-localization between YB-1 and Lin28 in their natural location, proximity ligation assays (PLA) were performed in HeLa cells expressing GFP-labeled Lin28 (Fig. 1d). LARP6, another RNA-binding protein without CSD, is used as a control because Lin28 and LARP6 share a very similar spatial distribution, being present both in the nucleolus and in the cytoplasm. The PLA signal indicating the colocalization of endogenous YB-1 and LARP6-GFP or GFP alone increases with the expression level, which is expected since the occurrence of having a GFP nearby YB-1 in the cytoplasm should increase. However, with Lin28-GFP, the colocalization signal is more intense for similar levels of expression which again reflects a possible co-localization between YB-1 and Lin28 in a cellular context.

**NMR analysis of the interaction of Lin28 with YB-1 in the presence of single-stranded nucleic acids (RNA, ssDNA).** CSDs are known to bind similarly and cooperatively to single-stranded nucleic acids, mRNA and ssDNA, whatever in bacteria[33] and mammals[34]. Given the homology between the CSDs of YB-1 and Lin28 (Supplementary Fig. S1d), a cooperative association to mRNA mediated by their common CSDs could provide a good basis for their interaction in cells. To test this hypothesis, we expressed two truncated forms of Lin28, Lin28-N-ter and Lin28-C-ter. Lin28-N-ter comprises the CSD and the positively charged linker domain that separates the CSD from its two C-terminal CCHC-type zinc knuckle domains (ZDK). Lin28-C-ter contains only the Lin28 ZDKs. When these truncated forms were used as baits on the microtubules to probe the co-localization with YB-1, only the N-terminal part comprising the CSD makes it possible to bring YB-1 on microtubules, which indicates that the CSD plays a preponderant role in the co-localization between YB-1 and Lin28 (Supplementary Fig. S2b).

Gel mobility shift assays are also in agreement with a mixing of Lin28-CSD with YB-1-CSD-rich ssDNA complex (Fig. 2c). Lin28 decreases the electrophoretic mobility of YB-1-rich ssDNA (20-nt long poly(C) ssDNA). A similar result was obtained with mRNA (Supplementary Fig. S3a). In contrast, the RRMs of TDP-43 or FUS lead to the appearance of ribonucleoprotein complexes of distinct electrophoretic mobility as if FUS or TDP-43-rich ssDNA coexists with YB-1-rich DNA without mixing (Supplementary Fig. S3a).

We then investigated by NMR spectroscopy the structural basis of the YB-1 and Lin28 binding to short single-stranded nucleic acids. As full length Lin28 is not sufficiently soluble to be amenable to NMR spectroscopy, we analyzed a truncated form comprising the CSD and the positively charged C-terminal linker (Lin28-CSD, aa 32–136). In addition, again because of solubility issues, part of the unstructured N-terminal region had to be removed (Supplementary Fig. S1a). In the case of YB-1, YB-1-CSD (aa 1–180) is the longest truncated form amenable to NMR that comprises the CSD and part of the positively charged CTD[7].

The NMR data of Lin28 interacting with let-7 has already been published but not that of free Lin28[35]. The truncated form used here, Lin28-CSD, therefore made it possible to obtain $^1$H-$^{15}$N

HSQC spectra of Lin28 free state and in complex with nucleic acids (10 nt-long Poly(C) DNA, Fig. 2a). A comparative analysis of the binding of Lin28- and YB-1-CSD to ssDNA indicates their similar binding to single-stranded nucleic acids (Supplementary Fig. S3b and Fig. 2b). Indeed, the conserved residues known to interact with nucleic acids in the CSD display similar chemical perturbations (CSPs) for Lin28 and YB-1 in the presence of nucleic acids (Supplementary Fig. S3b, c). In addition, NMR peak perturbations were observed for some Lin28 and YB-1 residues in loop 3 which are not involved in direct contact with nucleic acids[36], most probably due to structural arrangements upon binding to nucleic acids as previously described for YB-1[7].

To decipher the structural basis of the cooperative association of YB-1 and Lin28 in the presence of long single-stranded nucleic acids, we then used a 20 nt-long Poly(C) ssDNA that could accept at least two CSDs. As consequence, $^{15}$N-Lin28-CSD peak intensities drop in comparison to 10 nt-long ssDNA presence, due to the larger size of the complexes (Supplementary Fig. S3d). When $^{15}$N-Lin28-CSD interacts with 20 nt-long ssDNA at increasing concentrations of YB-1-CSD, the peak intensities drop again for the same reason, which is not observed when TDP-43 RRM2 was used instead of YB-1-CSD under the same conditions (Fig. 2d, Supplementary Fig. S3e). In addition, the decrease in the heights of peaks occurred mostly for residues located in the β-barrel structure, which is indicative of a tight packing of CSDs along ssDNA. CSP analysis revealed many residues that may be involved in the cooperative association of Lin28 and YB-1 to ssDNA (Fig. 3a, b). Residues of the β-barrel that might be involved in the binding to nucleic acids such as E105 were no longer considered (Fig. 3b, Supplementary Fig. S3a). A similar NMR analysis was also carried out for $^{15}$N-YB-1-CSD upon the addition of unlabeled Lin28-CSD or YB-1-CSD in the presence of 20 nt Poly(C) RNA and ssDNA (Fig. 3b, Supplementary Fig. S3f). CSPs were again detected in similar regions than those identified in Lin28-CSD such as in loop 3 (R85 and S86 for Lin28, S102 for YB-1), at the C-terminal end of β-sheet 3 (Q76 for Lin28 and Q88 for YB-1) but also in the CTD (G129, G135 for YB-1 and to a lesser extent G114, V115 for Lin28) highlighting their possible contribution to the cooperative association of Lin28 and YB-1 to single-stranded nucleic acids.

**Microtubule bench assays reveal critical residues for the mixing between YB-1 and Lin28 in cells.** After the identification of Lin28-CSD residues displaying CSPs on NMR spectra upon the mixing with YB-1 on ssDNA, the relevance of their respective contribution was assessed in a cellular context using microtubules as intracellular nanoplatforms. Conserved residues located in the folded β-barrel that bind to mRNA were discarded, except F47 that was used as an RNA-binding residue control. In addition, given that Lin28 and YB-1 display a similar cooperative binding to RNA[7,24], we targeted Lin28 domains, for which CSPs were also detected in the reciprocal YB-1 residues. Similarly, at the C-terminal end of β-sheet 3, Q76/S77 for Lin28 and Q88/T89 for YB-1 display CSPs when multimerization takes place (Fig. 3b), possibly revealing their contribution to the cooperative binding of the cold-shock domain to nucleic acids. We also considered residues such as G114/V115 at the beginning of CTD and two positively charged residues R122/R123 that may contribute to the electrostatic bridging of the Lin28 CTD to consecutive CSD. In total, the mixing/demixing between YB-1 and 18 full length Lin28 mutants was measured in HeLa cells. As controls of the micro-tubule bench efficiency for probing the mixing of RBPs with YB-1, we observed and measured the perfect mixing of YB-1 with itself and the strong demixing with G3BP-1 (Fig. 4a). In contrast, Lin28 and YB-1 are mixing well, even if to lesser extent than YB-1

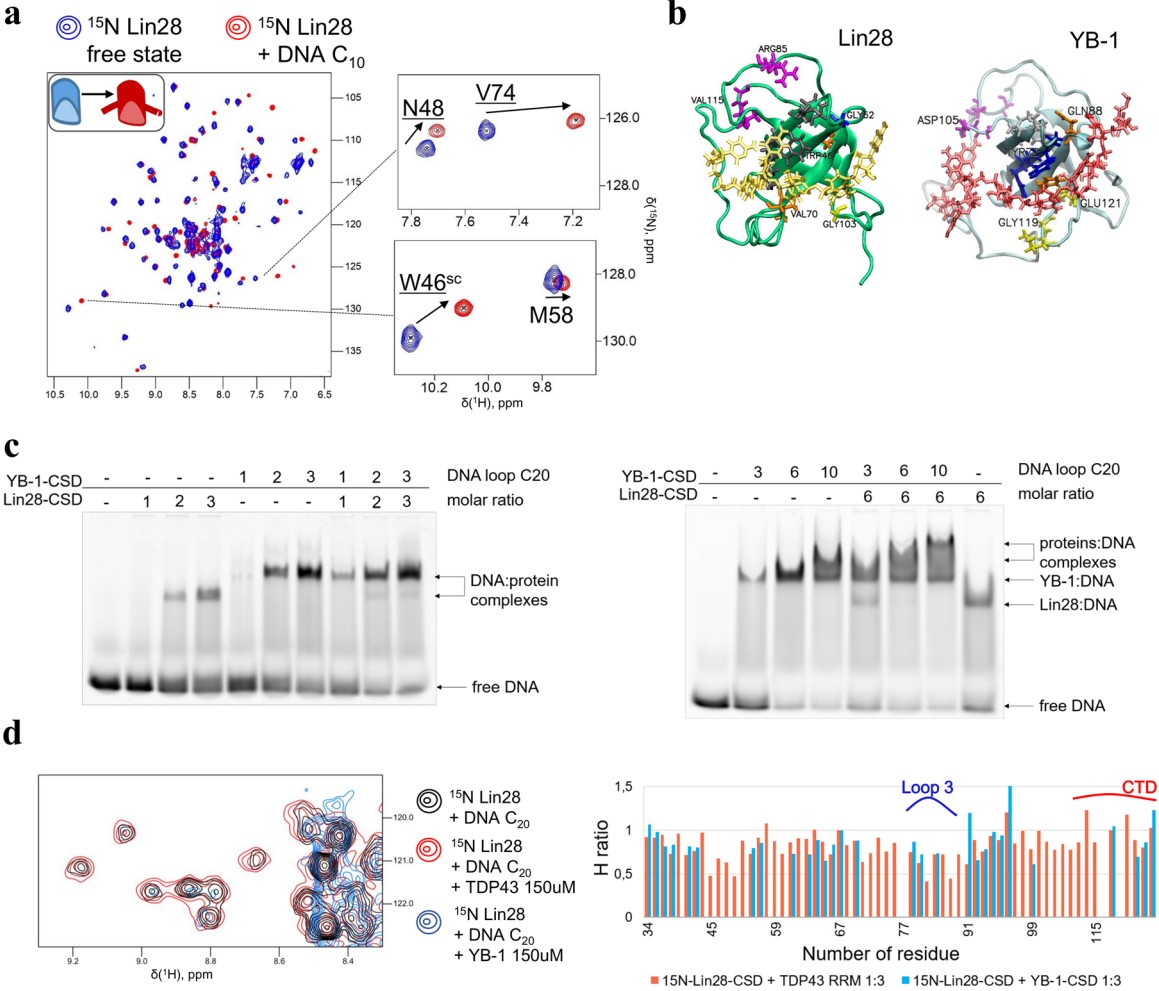

**Fig. 2 NMR analysis of the mixing of Lin28 and YB-1 on single-stranded nucleic acids. a** Two-dimensional $^1$H-$^{15}$N HSQC spectra of Lin28-CSD (32–136 aa.) in free state or in presence of 10 nt-long poly(C) oligonucleotides. **b** Lin28-CSD and YB-1-CSD residues perturbed upon binding to 10-nt-long ssDNA (in gold and red, respectively). The residues are colored differently depending on their location: gray - β-sheet 1, blue - β-sheet 2, orange - β-sheet 3, yellow - loop 4, violet – loop 3 and CTD. **c** Gel mobility shift assays of a 20-nt long Poly(C) DNA in the presence of YB-1-CSD and Lin28-CSD. The protein/DNA molar ratios are indicated. Proteins were premixed in a buffer solution containing Tris 20 mM, pH 7.6, NaCl 40 mM, DTT 0.5 mM, then DNA was added at room temperature for 30 min. **d** Left panel: Two-dimensional $^1$H-$^{15}$N HSQC spectra of Lin28-CSD in the presence of 20 nt-long ssDNA and TDP-43 RRM2 or YB-1-CSD in excess. Right panel: Height ratio of Lin28-CSD resonances in presence of 20 nt-long poly(C) DNA with/without TDP-43 RRM-2 or YB-1-CSD.

with itself (Fig. 4a). With Lin28 mutants, we identified 6 mutations that impair the YB-1/Lin28 mixing (Fig. 4b, c and Supplementary Fig. S4a, b). We first noticed that the mutation F47A significantly alters the mixing of Lin28 with YB-1 which confirms the essential role of mRNA in the interaction between YB-1 and Lin28. It might also be the case for K98/K99, two lysine residues located near nucleic acids in the Lin28:RNA complex (Fig. 4d) that may be engaged in electrostatic interactions with nucleic acids. We also noticed that mutations in the CTD, R122A/R123A, in loop 3, R85A, and at the end of the β-sheet 3, Q76A/S77A, are also critical for the mixing between YB-1 and Lin28 (Fig. 4b). As both Q76 and R85 are followed by a serine residue, we paid a particular attention to them owing to a putative role of Lin28 phosphorylation in translation control. However, their mutations into alanine, S77A and S86A, and into glutamic acid, S77E, did not significantly impair the mixing between YB-1 and Lin28 in both cases (Fig. 4c, Supplementary Fig. S4a, b). S77 and S86 phosphorylation then may not play a structural role but we cannot exclude the recruitment of additional factors in cells or a long-range structural transition that cannot be mimicked by alanine or glutamic acid mutation.

**Structural basis of the cooperative binding of YB-1 and Lin28 to RNA**. To understand the role of R85 in loop 3, Q76/S77 at the end of the β-sheet 3, and CTD residues that do not interact with RNA, we took advantage of the structure of the YB-1 trimer formed in the presence of Poly(C) RNA that we have recently determined in a previous study by combining molecular dynamics and NMR data[7]. The YB-1 trimer model is a good basis to explore the cooperative association of CSD proteins to mRNA by molecular dynamics, even if the relevance of this model in a cellular context remains to be demonstrated. Interestingly, we found that, unlike the RRM of TDP-43, Lin28-CSD is able to replace YB-1 in the trimer while preserving the binding of adjacent YB-1-CSD to RNA (Fig. 5a, b, Supplementary Fig. S5a). In addition, molecular dynamics indicates that Lin28 forms a stable homotrimer on RNA on its own (Supplementary Fig. S5a). By analyzing the molecular dynamics data for the homo- or heterotrimers with YB-1, we noticed that Q76 and S77 interact strongly with RNA at the Lin28/YB-1 or Lin28/Lin28 interface (−104.25 and −72.46 kJ/mol, respectively) but very poorly at the Lin28/RRM interface (−6.2 kJ/mol), thus contributing to the cooperative binding of CSD to RNA (Fig. 5b). A critical step to generate stable trimers also relies on the

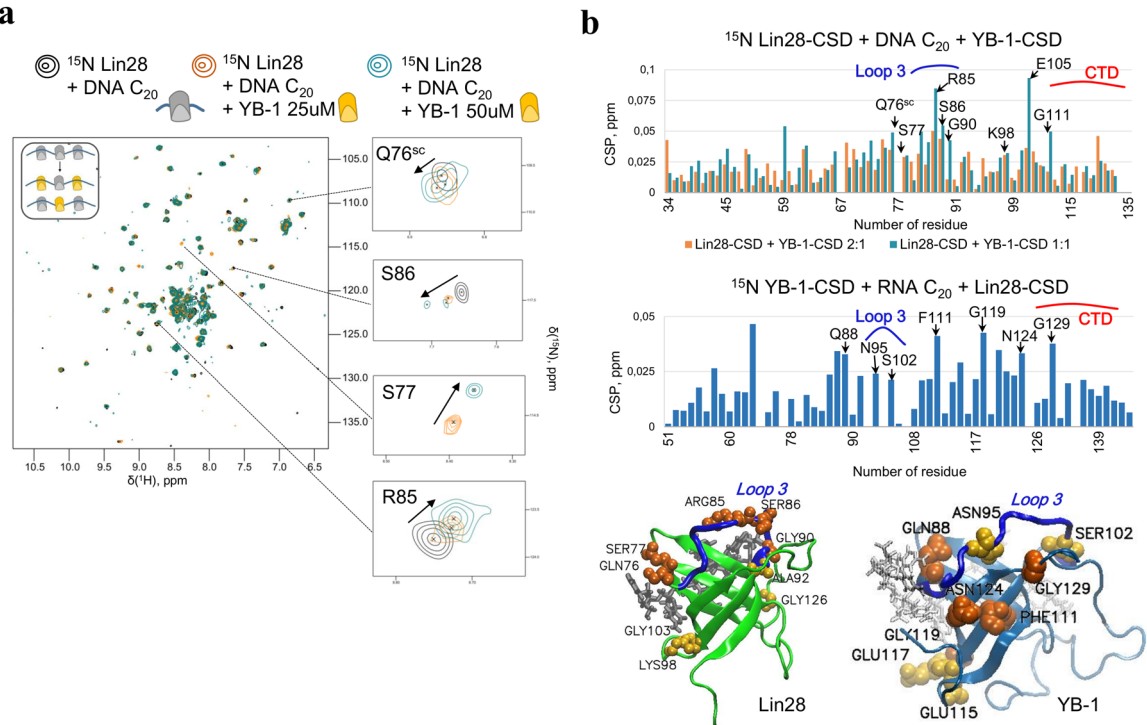

**Fig. 3 Identification of Lin28 residues possibly involved in the cooperative association of Lin28 and YB-1 to ssDNA and RNA. a** Two-dimensional $^1$H-$^{15}$N HSQC spectra of Lin28-CSD interacting with 20 nt-long poly(C) oligonucleotides in the presence of increasing concentrations of YB-1-CSD, as indicated. **b** Upper panel: CSPs of $^{15}$N-Lin28-CSD or $^{15}$N YB-1-CSD interacting with 20 nt-long poly(C) oligonucleotides in the presence of YB-1-CSD and Lin28-CSD, respectively. Lower panel: View on YB-1 and Lin28 CSDs showing the residues experiencing CSPs.

positively charged CTD that needs to be directed towards nucleic acid backbone located nearby the next CSD, therefore constituting the bridging system shared by both YB-1 and Lin28 (CSD plus a positively charged CTD). This notion is in agreement with the significant demixing observed after neutralizing two positively charged arginine residues in the Lin28 CTD (R122A/R123A) that should perturb the electrostatic bridging by CTD. We then asked whether specific residues enable the proper orientation of the CTD, notably residues located in loop 3 (Fig. 5a). R85/S86 by themselves may not be directly involved (Supplementary Table 1) but their mutations into alanine residues reduce the dynamics of loop 3 and modify its interactions with CTD residues (Supplementary Fig. S5b, c). Molecular dynamics analysis rather suggests intramolecular interactions between K88-E91 in the loop 3 with I118-R122 located at the beginning of the CTD (Supplementary Table 1, Fig. 5a). We then considered whether G119/S120 residues, in the middle of this CTD sequence, could participate to the mixing between YB-1 and Lin28. G119/S120 had not been selected initially since they were not visible in the NMR spectra of Lin28 in the presence of nucleic acids. However, the resonance peaks of G135/S136 in YB-1 corresponding to conserved residues G119/S120 in Lin28, reappeared or shifted, respectively, in the presence of Lin28. According to molecular dynamics, G135/S136 could also be involved in YB-1 intramolecular interaction between loop 3 and CTD (Supplementary Table S1). We then used the microtubules as nanoplatforms to probe experimentally the putative role of G119/S120 in the mixing of Lin28 with YB-1 and itself. The results indicate that the double mutation G119A/S120A induces a significant demixing in cells which supports the critical role of the intramolecular interaction between Lin28 loop 3 and the beginning of CTD in the mixing between YB-1 and Lin28 (Fig. 5c). The results therefore point towards an intramolecular interaction responsible for the orientation of the CTD making possible the cooperative assembly of YB-1 and Lin28 along mRNA.

**Interplay between Lin28 and YB-1 in cultured cells**. Given the cooperative association of Lin28 and YB-1 in the presence of RNA and having identified mutations that interfere with their mixing, we examined the relevance of this interaction in the cytoplasm of HeLa cells. We started by considering whether the co-localization between YB-1 and Lin28 that we observed on microtubules could be detected by immuno-precipitation. In HEK293 cells expressing Lin28-GFP, the presence of endogenous YB-1 was detected in Lin28-GFP-immunoprecipitates without but not with RNAse treatment, in agreement with a cooperative association of Lin28 and YB-1 in the presence of mRNA (Fig. 6a). We also selected two mutants to perform the immunoprecipitation assays, Lin28-RS (R85A/S86A) that disrupts the intramolecular interaction responsible for orienting the CTD, and Lin28-QS (Q76A/S77A) that impairs the cooperative binding of the CSD to RNA (Fig. 6a). We preferred to use the double mutations that include the serine residues, S77 and S86, to prevent any bias that would be induced by their differential phosphorylation in cells. When Lin28-RS and Lin28-QS were used to pull down YB-1 instead of wild type Lin28, a decreased amount of YB-1 was detected in the IP fraction compared to YB-1 pulled down with wild type Lin28-GFP, as expected from an impaired cooperative association of Lin28-RS and-QS with YB-1 in the presence of mRNA (Fig. 6a).

We then consider a possible cooperative association of Lin28 and YB-1 to stress granules that are liquid-like compartments in which mRNAs are gathered to reorganize the translational response to stress. Here, stress granules were assembled after the exposure of HeLa cells to arsenite that triggers the rapid phosphorylation of the initiation factor eIF2A[37], thus allowing the dissociation of polysomes and thereby the subsequent formation of stress granules. To mimic the experiments performed on microtubules, YB-1 labeled with HA-tag (in red) and three GFP-labeled RBPs, G3BP1, a core stress granule protein, LARP6, and Lin28 were expressed in

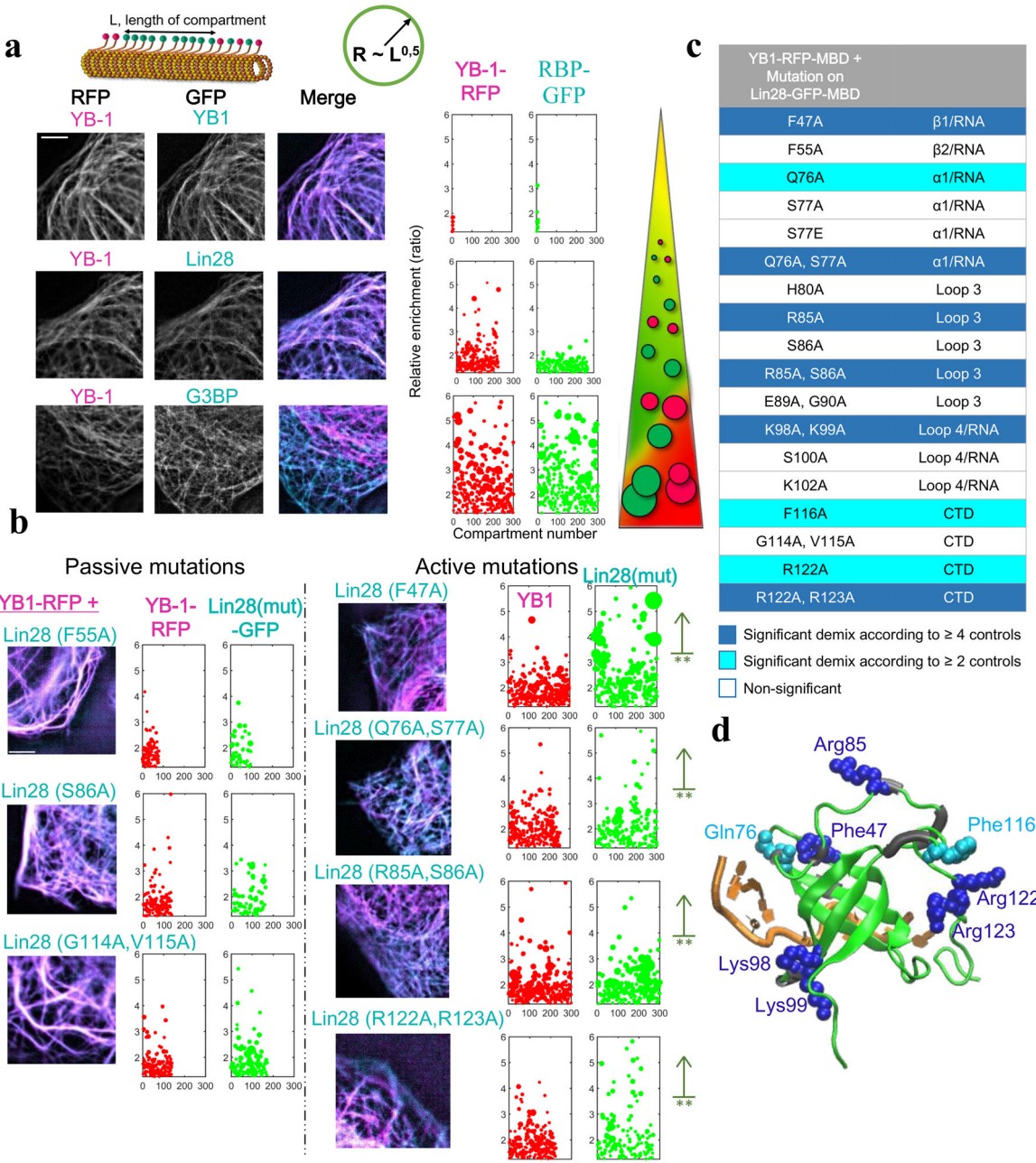

**Fig. 4 The microtubule bench identifies Lin28 residues involved in the mixing of YB-1 and Lin28 in cells. a** Left-panel: Representative images of the mixing/demixing of two full length RBPs along microtubules in HeLa cells. Right panel: Plot of the length and enrichment of RFP or GFP compartments measured along the microtubule network in cells (>7 mm per condition, see Methods for details). Sphere radii in the graphs are proportional to square root of length of the corresponding compartments. Scale bar: 5 μm. **b** Same as **a** with representative images of the mixing/demixing between Lin28 mutants and YB-1 along microtubules in HeLa cells. **\****p* < 0.01, Student's test with two tails compared with wild type Lin28 (see Methods for details). Scale bar: 5 μm. **c** Identification of Lin28 mutants leading to a significant demixing with YB-1 compared to wild type Lin28 and passive mutations. Both Student's and Kolmogorov–Smirnov's tests were used to compare all the mutants with each other (see Supplementary Fig. S4b). Non-significant demixing, white. Significant demixing, cyan (according to ≥ 2 controls) or blue (according to ≥ 4 controls). **d** View of residues leading to YB-1 and Lin28 demixing when mutated into alanine residues.

HeLa cells prior to arsenite exposure (HeLa cells do not express Lin28 endogenously). In contrast with Lin28-GFP, stress granules in cells expressing LARP6 or G3BP-1 appeared with a greenish color, which indicates a reduced presence of YB-1 in these stress granules (Fig. 6b). Measurements of the relative enrichment of YB-1 further suggests a better association of YB-1-HA with Lin28 in stress granules compared with G3BP-1 and LARP6 (Fig. 6b).

The putative mixing of YB-1/Lin28 in stress granules was then probed with endogenous YB-1 and mixing-deficient Lin28 mutants, Lin28-RS and -QS. Lin28-DE (D33A/E34A) was used as passive double mutation (E34 is not affected as seen in NMR spectra and the N-terminal part of Lin28-CSD does not display major peak perturbance in the presence of YB-1). As control that these mutations do not alter dramatically the binding of Lin28 to

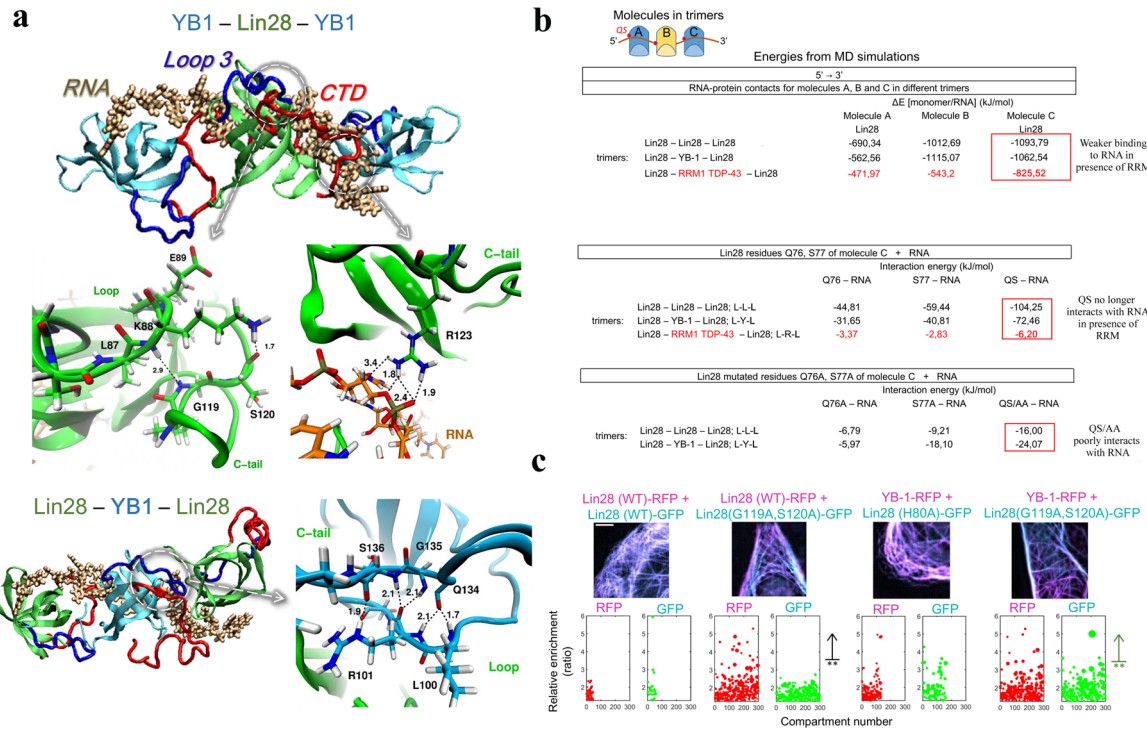

**Fig. 5 Molecular Dynamics (MD) data for Lin28 and YB-1 heterotrimers indicates the mechanism behind the mixing between YB-1 and Lin28 in the presence of RNA. a** MD structures of YB-1 and Lin28 hetero-and homotypic trimer formed in the presence of 16 nt-long Poly(C) RNA (see Materials and Methods for details). Zooms in on the interaction between CSD loop 3 and CTD that controls the cooperative association to RNA. **b** Interaction energies at the protein-protein and protein-RNA levels for indicated homo- and heterotrimers (ABC) interacting with 16-nt long Poly(C) RNA. Upper panel: global RNA-protein interactions for each protein, A, B, and C. The interaction energy of protein A with RNA is low since protein A interacts with less nucleotides than proteins B and C in the presence of the 16-nt-long RNA. When RRM1 of TDP-43 is located in the middle of the trimer (as molecule B), it significantly reduces the interaction of flanking proteins, A and C, with RNA. Middle panel: Interaction energies of Q76 and S77 of protein C with RNA for indicated homo-or hetero-trimers. Q76 and S77 of protein C are located at the interface between protein B and C. We noticed that Q76 and S77 poorly interact with RNA when protein B is TDP-43 RRM1 compared to YB-1 or Lin28. Lower panel: energies of interaction between Q76, S77 being mutated into alanines, with RNA in indicated trimers. In comparison to wild type, the contact of mutated Lin28 residues to RNA decreases dramatically. Energies were averaged over 200 ns of MD simulation (for RRM1 of TDP-43–10 ns were sufficient to observe a reduced interaction) and values are reported in kJ/mol with variant of fluctuations being ± 0.4 kJ/mol. **c** Representative images of the mixing/demixing Lin28 mutants and YB-1 along microtubules in HeLa cells. The relative enrichment of Lin28 and YB-1 compartments was measured as described in Methods. G119A and S120A mutations lead to a marked demixing between YB-1 and Lin28. H80A is a negative control. Scale bar: 5 μm.

mRNA, we confirmed that the Lin28 mutants, used as baits, can bring mRNA on microtubules in cells (Supplementary Fig. S2a). In addition, using proximity ligation assays, we found that Lin28-QS, -RS and -DE still colocalize with endogenous YB-1 in cells (Supplementary Fig. S6a). However, when YB-1 was used as bait to bring Lin28 on microtubules, Lin28-QS and -RS mutants formed distinct compartments on microtubules while wild type Lin28 and Lin28-DE displayed a continuous distribution along microtubules (Supplementary Fig. S2c). Altogether these results show that Lin28-QS and -RS mutations induce subtle changes that modify the mixing/demixing between Lin28 and YB-1 but without impairing significantly their affinity for mRNA and without disrupting totally the interaction between Lin28 and YB-1. Finally, we noticed that the spatial distribution of the wild type and Lin28 mutants was very similar (Supplementary Fig. S6c). In addition, using CellProfiler software[38], we controlled that the endogenous expression level of YB-1 did not significantly change after the expression of Lin28 (Supplementary Fig. S6d).

After the validation of the Lin28-RS and -QS mutants, we then measured the enrichment of YB-1 and Lin28 in stress granules in arsenite-treated cells automatically at the single cell level using CellProfiler (Fig. 6c). We noticed a linear relationship between the enrichments of YB-1 and Lin28 in stress granules. This

behavior is not specific to Lin28 and YB-1 since the enrichment of LARP6 in stress granules also increases with that of YB-1, though to a lesser extent than with Lin28 (Supplementary Fig. S6b), most probably because denser stress granules may recruit more RBPs. To probe the interplay between Lin28 and YB-1 in stress granules, we then decreased the level of YB-1 by siRNA. If Lin28, which concentrates in stress granules, stabilizes the presence of YB-1 inside them, YB-1 level in stress granules should decrease to a lesser extent than in the cytoplasm (Fig. 6c). After reducing the expression of YB-1 by siRNA, we detected a modest but significant increase in the relative enrichment of YB-1 in stress granules. Such increase is still significant when Lin28-DE (control mutant) is expressed but not for Lin28-RS and -QS mutants (Fig. 6c). Lin28-RS and -QS mutants therefore poorly cooperate with YB-1 for its association with mRNA-rich stress granules.

In addition to data obtained with stress granules, immuno-precipitation of wild-type Lin28 or Lin28-RS with consequent RT-qPCR indicates a higher affinity of wild type Lin28 than Lin28-RS for cellular mRNAs (Fig. 6d, Lin28-QS was not considered in this analysis because Q76/S77 interact with RNA, Fig. 5b). Given the altered cooperative association of Lin28-RS and YB-1 to mRNA, YB-1 may become a competitor for the binding of Lin28-RS to mRNA. In agreement with this

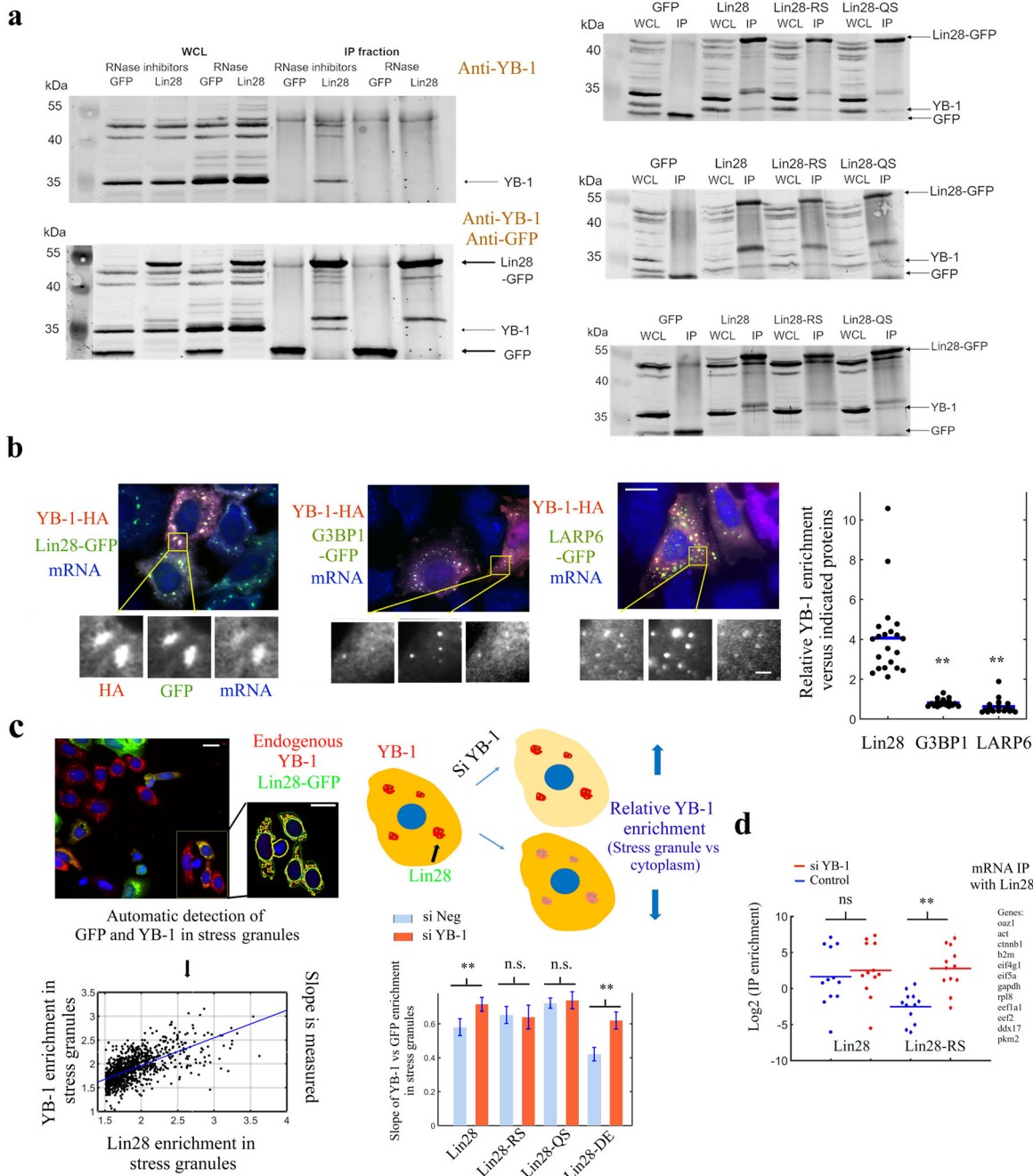

hypothesis, decreasing the expression of YB-1 by siRNA increases significantly the affinity of Lin28-RS for mRNA. For wild type Lin28, decreased levels of YB-1 have no significant effect on its affinity for mRNA probably because, in contrast with Lin28-RS, Lin28 binds to YB-1-rich mRNA, apart from binding to YB-1-free mRNAs on its own.

**First hints of possible functions related to the Lin28-YB-1 co-association to mRNPs.** We then considered whether Lin28 and its cooperative binding with YB-1 to mRNAs affects cell proliferation. After the expression of Lin28-GFP, the proliferation rate of HeLa cells decreases, as measured by BrdU staining at the single cell level (Supplementary Fig. S7b). However, this is not specific to Lin28. Many RBPs such as LARP6, used here as control, decrease cell proliferation when overexpressed in cells, but, unlike cells

expressing LARP6, the proliferation rate can be partially restored in cells that express Lin28 when YB-1 expression is reduced. The negative control on cell proliferation exerted by Lin28 expression is thus dependent on endogenous YB-1. For the mutants, Lin28-RS, -QS, we observed the same blocking phenotype regarding proliferation but less marked than wild type Lin28. However, when YB-1 levels are reduced, cells expressing Lin28-RS or -QS better recovered their proliferative status than with wild type Lin28 (Supplementary Fig. S7c), which is consistent with a better mixing between Lin28 than Lin28-RS and –QS with the remaining YB-1 pool that still partially represses cellular proliferation.

To further explore the hypothesis of a functional interplay between Lin28 and YB-1, we then used the fact that Lin28-GFP expression promotes neurite growth in a neuronal cultured cell line (here NSC-34) in vitro, which is an established Lin28 phenotype[19] at least partly independent of let-7[39]. In agreement

**Fig. 6 Interplay between Lin28 and YB-1 in cells. a** Left panel: Western blot analysis of whole cell lysate (WCL) and anti-GFP immunoprecipitates (IP) of HEK293 cells expressing GFP or Lin28- GFP with or without RNase treatment. (Upper Gel: only anti-YB-1 antibody. Lower Gel: Anti-GFP and anti-YB-1 antibodies). Right panel: Co-immunoprecipitation of endogenous YB-1 with wild-type Lin28, Lin28-RS or –QS. IP fractions show a smaller amount of YB-1 co-precipitating with the two Lin-28 mutants comparing to wild type Lin28. Three independent experiments are shown. Anti-GFP and anti-YB-1 antibodies were used. **b** Left panel: Representative images of stress granules in arsenite-treated HeLa cells expressing YB-1-HA and GFP-labeled RBPs as indicated. Zooms in on stress granules show the relative enrichment of YB-1 and GFP-labeled RBPs and the presence of mRNA (in situ hybridization with oligo-d(T) probes). Right panel: Ratio of the enrichment of YB-1 versus indicated RBPs in stress granules for similar expression levels (see Methods). $n = 21$. $**p <$ 0.01, $t$-test with two tails versus Lin28. Scale bars: 15 µm and 2 µm (higher magnification, lower panel). **c** Left panel: Stress granules are detected in arsenite-treated HeLa cells expressing Lin28-GFP using CellProfiler. Anti-YB-1 (Red). The enrichment of YB-1 and Lin28 in stress granules is then measured and plotted to quantify the slope of their relative enrichments. Upper right panel: Scheme representing the consequences of decreasing YB-1 expression on the relative enrichment of YB-1 in stress granules. Lower right panel: Slopes of YB-1 versus wild type or mutant GFP-labeled Lin28 enrichments in stress granules after decreasing or not YB-1 levels are represented. Ratios are given with 95 % confidence bounds. Scale bar: 15 µm. **d** RT-qPCR analysis of the mRNA content of the anti-GFP immunoprecipitates (IP) in HEK293 cells expressing Lin28-GFP (RS mutant or wild type) with (red) or without (blue) decreasing endogenous YB-1 levels with siRNA. mRNA was extracted from the IP fraction using the standard protocol (see Materials and Methods), then RT-PCR measurements were performed to reveal the presence of mRNAs encoding for genes indicated on right panel. The genes were chosen according to their abundance in HEK cells to avoid imprecise measurements. We noticed that the enrichment of mRNAs in the IP fraction increased significantly when YB-1 levels were decreased for the Lin28-RS mutant compared to wild type Lin28. YB-1 is therefore a competitor for the binding of Lin28-RS to mRNA most probably because of an impaired cooperative association with YB-1 to mRNA.

with the reported phenotype, the expression of Lin28-GFP clearly promotes neurite extensions of neuronal cells. In contrast, the expression of other RNA-binding proteins such as HuR, G3BP1, LARP6 and TDP-43 does not allow to reproduce this phenotype while YB-1 prevents the formation of neuritic extensions (Supplementary Fig. S7b). Therefore, YB-1 and Lin28 seem to act differently on the formation of neurites. However, when the expression of YB-1 is decreased by siRNA, the extension of neurites is again reduced, suggesting that an optimal level of YB-1 is required for an efficient axon formation. Interestingly, in cells expressing Lin28-GFP, decreasing YB-1 expression has a dramatic impact on the capacity of cells to form neuritic extensions. In contrast, Lin28-RS cannot increase the occurrence of axonal extensions, whatever the YB-1 levels (Supplementary Fig. S7b). These results suggest that Lin28 promotes axon formation in a YB-1-dependent manner that does not rely on an additive effect since YB-1 overexpression suppresses neuronal extensions.

## Discussion

YB-1 and Lin28 are two mRNA-binding proteins whose expressions are tightly controlled during organism development sharing similarities in their functions such as the processing of mRNPs during spermatogenesis[40,41] and embryogenesis[42,43]. The results provided here demonstrate that Lin28 co-localizes with YB-1-rich mRNPs, which opens the perspective of a global regulation of mRNA translation, independently of the let-7 pathway, through the cooperative association to mRNPs of the structurally similar YB-1 and Lin28 CSD[44]. The cooperative association of YB-1 and Lin28 in the presence of mRNA presents a high degree of specificity because, besides Lin28, only CSDE1 that has 5 CSDs (Fig. 1b, Supplementary Fig. 1a, b) and most probably CHSP1 and Lin28b, the paralogue of Lin28, may associate cooperatively with YB-1 to mRNA. The other RBPs without CSD tested in our study, G3BP1, FUS, LARP6, and HuR do not co-localize with YB-1-rich mRNPs, possibly due to the tight packing of CSDs along mRNA that may not be suitable for other RNA-binding domains (Figs. 2c and 4a, b).

**Structural basis of the mixing between Lin28 and YB-1**. We found that the interaction between Lin28 and YB-1 is not direct but based on a cooperative association of their CSD to mRNA (Figs. 1a, b and 5a, Supplementary Fig. S2b). This is not

surprising as very few direct interactions between RBPs have been revealed so far, apart from specific cases such as the NONO/SFPQ heterodimer. A cooperative binding to mRNA most probably constitute an important mechanism for the specific targeting of mRNA by RBPs[45] that cannot be predicted solely by the specific binding of RBPs to few nt-long sequences. To enable an efficient mixing between YB-1 and Lin28 in the presence of mRNA, the positively charged CTD that are present in both YB-1 and Lin28 must be oriented towards the sugar-phosphate backbone of mRNA strands interacting with an adjacent CSD. We identified that the deployment of both the YB-1 or Lin28 CTD relies on an intramolecular interaction between few residues located in the loop 3 and at the beginning of the CTD (Fig. 5a).

**Role of the zinc finger domain of Lin28?** In addition to the CSD, Lin28 and its paralogue Lin28b also have another structured domain, the C-terminal CCHC-type zinc knuckle domain (ZDK), which is not present in YB-1. In our model, the zinc finger domain of Lin28 has no room to bind to mRNA due to the tight packing of CSDs along mRNA (Figs. 5a and 7c). The zinc finger domain would be therefore free to interact with possible partners such as non-coding RNA or other biomolecules to direct mRNPs to specific compartments such as membranes[22]. Interestingly, Lin28 is associated to mRNAs at the endoplasmic reticulum surface, which explains the diffuse but partially perinuclear location of Lin28 in cells[22].

**Functional interplay between endogenous Lin28 and YB-1 in vivo**. To further explore the possible functional link between Lin28 and YB-1, we have considered the published data available regarding the endogenous expression of Lin28. Based on protein expression reflected by RNA-Seq data, which are more reliable than proteomic data, we measured the correlation of the expression level of Lin28 with other RBPs across 60 human tissues[46]. To perform this analysis, we sought all proteins appearing as "RNA-binding proteins" and selected only the most abundant ones which are expressed in all tissues to make a fair comparison with YB-1 (47 proteins in total, see Materials and Methods), among them are FUS, HuR (ELAVL1) and G3BP-1. Strikingly, Lin28 expression has the best correlation across tissues with YB-1 expression (Fig. 7a), mostly because Lin28 and YB-1 are highly expressed in testis. YB-1 and Lin28 may therefore interact cooperatively with the paternal mRNA. Similarly, during

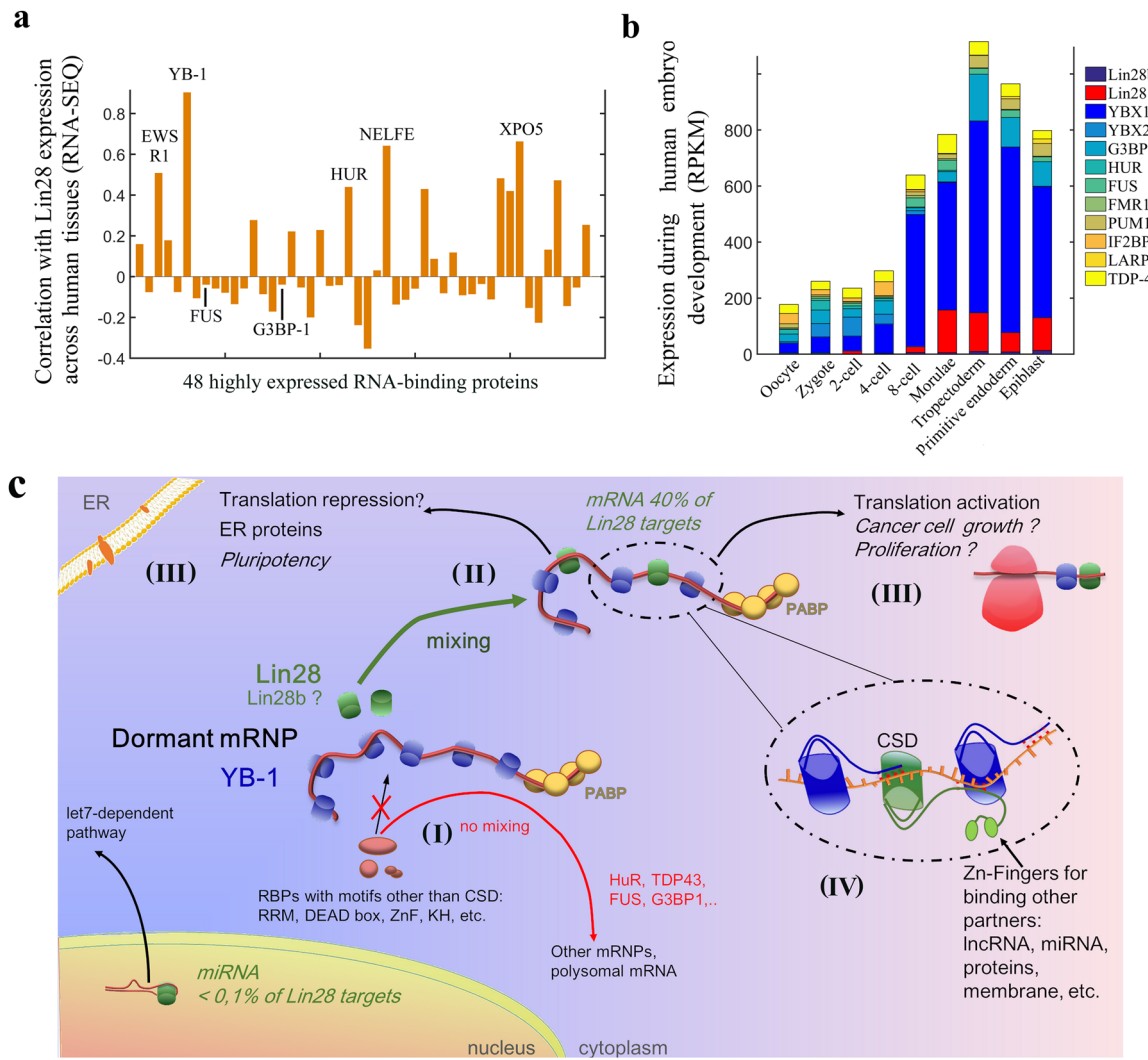

**Fig. 7 Putative functional consequences of a cooperative association of Lin28 to YB-rich mRNPs. a** Correlation (Pearson correlation coefficient) of the expression levels of Lin28 with 48 RBPs abundantly expressed in all human tissues[46] such as YB-1 based on RNA-Seq data. **b** Expression levels of YB-1 and YB-2, the two most expressed Y-box-binding proteins expressed in embryo, Lin28 and Lin28b, and other RBPs, as measured during different stages of embryogenesis[47]. **c** Schematic representation of the Lin28 possible functions in cells. (I) The demixing between YB-1 and most RBPs prevents the access of RBPs to YB-1-packaged mRNAs. (II) Lin28 can bind to YB-1-packaged mRNA thanks to its cold-shock domain. (III) Putative outcomes of the binding of Lin28 to mRNPs: activation of translation (glucose metabolism, etc.) or translation repression (membrane proteins, etc.). (IV) Zoom in on the structure leading to the cooperative binding of Lin28 and YB-1 to mRNA. The percentage of Lin28a RNA targets were obtained from published data[2].

embryonic development the management of mRNPs, their transport and their activation are finely regulated. We then analyzed the temporal expression of YB-1 and Lin28 using data obtained during embryogenesis at the single cell level[47]. The peak of Lin28 expression occurs during morulae and early blastocyst which correlates with that of YB-1, again pointing towards a YB-1-dependent function for Lin28 in vivo (Fig. 7b).

**mRNPs as critical players for reprogramming cells.** What could be the point for Lin28 to be directed to YB-1-packaged mRNPs? As noticed in a previous study in HeLa cells, a significant fraction of dormant mRNPs encodes for proteins involved in regulation of transcription[48]. In addition, dormant mRNPs are significantly less abundant than polysomal mRNAs encoding for housekeeping proteins[48]. The expression of key proteins associated to cell reprogramming can thus be turned off/on by the binding of an RNA-binding protein to mRNPs such as Lin28 that is not as abundant in

most tissues as YB-1, HuR, G3BP1 and others. Changing the expression level of dormant mRNPs could notably have major consequences in cell reprogramming linked to pluripotency and cancer. A cooperative association between YB-1 and Lin28 to mRNA may also control the translational response to stress. YB-1 and Lin28 are both components of stress granules that are liquid-phase compartments[49] in which some non-polysomal mRNAs[50] are gathered during environmental stress such as hypoxia, oxidative and genotoxic stresses, but also after viral infections[49].

In summary, the results of this study demonstrate a cooperative association of Lin28 and YB-1 in mRNPs which may indicate that Lin28 could reprogram mRNA translation in cells independently of let-7 through its association to YB-1-rich mRNPs (Fig. 7c). This model may also serve as a basis to explore the functional interplay between cold-shock proteins such as CSDE1, YB-1 and Lin28a/b and their role in pluripotency, cell proliferation, neurogenesis[31,51] and the plasticity of cancer cells allowing their resistance to chemotherapy.

# Methods

**Plasmid preparation and protein overexpression**. The Lin28-CSD sequence (32–136 aa., Supplementary Fig. S1a) was amplified from coding region of *H. sapiens* Lin28, cloned into pET28, with added restriction sites for NdeI/XhoI for further insertion into expression vector pET22b containing (His)$_6$-tag sequence. Chemically competent *E.coli* BL21(DE3) cells were transformed with obtained plasmid and grown in LB medium (for non-labeled proteins) or in minimal medium M9 with added $^{15}$NH$_4$Cl and/or $^{13}$C-glucose (for labeled proteins) at 37 °C. Induction of protein expression was performed by adding IPTG 1 mM at OD600 = 0.6. The cell culture was incubated for 3.5 h at 37 °C after induction, then harvested by centrifugation at 2000 × *g* at 4 °C for 20 min.

**Protein purification**. His-tagged Lin28-CSD was purified under native conditions. The cells were resuspended in lysis buffer 20 mM Tris-HCl, pH 7.6, 2 M NaCl, 10 mM imidazole, 0.5 mM DTT, 0.5 mM PMSF, protease inhibitors tablets (Roche), and sonicated. The suspension was centrifuged at 18,500 × *g* for 1 h at 8 °C, the supernatant was put for agitation with Ni-NTA agarose resin for 1 h at 8 °C, then loaded to column. The 5 wash steps were performed with buffer 20 mM Tris-HCl, pH 7.6, 0.5 mM DTT, 0.5 mM PMSF, gradient of NaCl 2 M–500 mM and imidazole 10–40 mM. Protein was eluted within 3 steps with similar buffer containing 250 mM imidazole. The eluted fraction was dialyzed against 20 mM potassium phosphate buffer, pH 7.0, 500 mM NaCl, 0.5 mM DTT, and concentrated with Amicon Ultra Centrifugal Filters (Merck). Purification of YB-1-CSD (1–180 aa., Supplementary Fig. S1a) was performed in similar way[7].

**Molecular dynamics (MD) simulations**. All Homo- and Hetero-trimer complexes from our experiments were considered for MD simulations. These include: Lin28:RNA(C16) homo-trimer complex, YB-1:RNA(C16) homo-trimer complex, (Lin28:YB-1:Lin28):RNA(C16) hetero-trimer complex, (YB-1:Lin28:YB-1):RNA(C16) hetero-trimer complex and (Lin28:RRM1-TDP43:Lin28):RNA(C16) hetero-trimer complex.

The starting coordinates for these complexes were based on the homo-trimer complex YB-1:RNA[7], recently published by our team, and the models were constructed by homology modeling. The monomer of Lin28 bound to RNA was taken from the X-ray structure of human Lin28A in complex with let-7f-1 micro RNA pre-element (PDB ID 5UDZ, resolution 2 Å)[52]. The protein sequence used was limited to the CSD and part of the CTD going from A32 to G136 (length 105 aa.). The 9 missing amino acids in the X-ray structure (from K127 to K135) were constructed by homology modeling using the trimer complex YB-1:RNA as reference. And, the monomer of TDP43 RRM1 bound to RNA (from T103 to P178) was taken from the NMR structure of TDP43 monomer in complex with UG-rich RNA (AUG12), PDB ID 4BS2[53].

All MD simulations were carried out using GROMACS software[54] version 2018.2 with the "all atom" force field amber ff03[55] with associated nucleic acid parameters and periodic boundary conditions. The protonation states of the residues were adjusted to the pH used in our experiments (pH=7). The systems were solvated in a 80 × 120 × 100 Å box of TIP3P[56] water. A [NaCl] of 300 mM was used and counter-ions were added to neutralize the system. Each system was first energy minimized using 5000 steps of steepest descent, then heated from 0 to 298 K at constant volume for 500 ps and equilibrated in the NPT ensemble at *p* = 1 atm for 500 ps which was followed by 200 ns of NPT production run (except for Lin28: RRM1-TDP43:Lin28:RNA trimer complex where we ran only 10 ns of MD, which was sufficient to observe the destabilization of the complex). The Velocity Rescaling[57] (with t = 0.1 ps) and Parrinello-Rahman[58] methods were used for temperature and pressure control, respectively. The equations of motion were propagated with the leap-frog algorithm[59] and the time step was Δt = 2 fs. The particle mesh Ewald (PME) method was used for electrostatic interactions, with grid spacing of 1.6 Å, a relative tolerance of $10^{-5}$, an interpolation order of 4 for long-range electrostatics, and a cutoff of 14 Å together with a 12 Å switching threshold for LJ interactions. Bonds involving hydrogen were constrained by LINCS[60]. An energy decomposition analysis was performed to compute (I) intermolecular interactions at the protein-RNA binding interface and (II) intramolecular interactions between Loop 3 and the beginning of CTD during MD simulation to assess potential contributions of local interactions to stability. Energies reported in the manuscript are averaged over the MD simulation (Fig. 5b, Supplementary Table 1).

**NMR experiments**. NMR experiments were performed on NMR spectrometer Bruker AVIII HD 600 MHz with triple-resonance cryoprobe, in 1.7 mm capillary tubes using 60 μl of sample with 50 μM of $^{15}$N-labeled protein (50 mM potassium phosphate buffer, pH 7.0). Spectra for Lin28-CSD, YB-1-CSD free-state and bound to oligo-DNA C10 were obtained at 298 K (YB-1 spectra were already published[7]). Experiments with Lin28-CSD and YB-1-CSD bound to oligo-DNA/RNA C20 as well as mixes of proteins, including TDP43 (176–277aa.) (molar ratio protein(s): DNA 1:1) were performed at 303 K. The chemical shifts and peak intensities obtained from NMR spectra presented in this article are shown in Supplementary Data 2.

Lin28-CSD resonance assignments were performed using 2D $^1$H-$^{15}$N HSQC, 3D HNCA, 3D HNCO, 3D $^1$H-$^{15}$N NOESY-HSQC spectra, and the data kindly provided by the laboratory of Dr. Piotr Sliz, Harvard Medical School, USA. In total for Lin28-CSD construct 67% of non-proline residues were assigned. Several residues from CTD are visible just in presence of oligo-DNA C20 while it is in excess. The assignment of YB-1-CSD (1–180) was performed previously[7].

As an external reference 2-dimethyl-2-silapentane-5-sulfonic acid diluted in D$_2$O was used for chemical shift referencing. TopSpin 3.5pl7 (Bruker) and CcpNmr Analysis 2.4.1 software was used for data processing and analysis, respectively (Figs. 2 and 3, Supplementary Fig. S3).

**Electrophoretic mobility shift assay (EMSA)**. For EMSA experiments truncated forms of human YB-1 (CSD, 1–180aa.), Lin28a (CSD, 32–136aa.), TDP43 (RRM 1–2, 101–277aa.), FUS (RRM, 165–385aa.) were used. For experiments with circular phage ssDNA M13mp18 (Biolabs), agarose gels 0.8% stained with EtBr were used, buffer TAE 1X, at 5 V/cm for 40 min. Concentration of DNA 0.07 μM, for proteins – 0.45, 1.41, 4 μM for points 53, 17 and 6 nt per protein, respectively (Supplementary Fig. S3a). The binding buffer contained Tris 20 mM, pH 7.6, NaCl 40 mM, DTT 0.5 mM, the reaction was hold at room temperature for 30 min. For reactions in presence of 2 proteins, DNA was added as last step. For DNA stem loop (GT)24 + 16 bp, (C)$_{45}$ + 16 bp and oligo-ssDNA C20-[Cy3], the DNA concentration was 0.4 μM, for proteinsit is indicated on the figures (Fig. 2c, Supplementary Fig. S3a). The products of binding reaction were separated in acrylamide gel 8%, TAE 1X. For stem loop experiments the gel was stained with EtBr 0.5 μg/ml after running, for ssDNA C20-[Cy3] the fluorescence was detected with Amersham Typhoon bioimager with 532 nm excitation laser, 570 nm emission filter.

**Cell culture**. Mouse Motor Neuron-Like Hybrid Cell Line, NSC34, and Human Embryonic Kidney 293 cell line, HEK293, and HeLa cell line (American Type Culture Collection, USA) were cultured at 37 °C in a humidified atmosphere with 5% CO$_2$ and maintained in the high glucose formulation of DMEM (Life Technologies) supplemented with penicillin G 100 U/ml, streptomycin 100 μg/ml and fetal bovine serum (FBS) 5% (10% for HeLa cells; Thermofisher). The cells at confluence 10$^6$ were plated in 4/24-well plates and were transiently transfected with plasmids, carrying the studied protein gene, at a final concentration of 1 μg using lipofectamine 2000 (Thermofisher) transfection reagent for 24–72 h, depending on experiment, according to the manufacturer's instructions. Before each experiment using NSC34, the cell differentiation was induced by addition of retinoic acid (1 μl per 1 ml of medium) and incubation for 72 h. For microscopy samples preparation, cells were grown on glass bottom dishes (MatKek Corporation).

YB-1 siRNA [794 sense 5'-(CCACGCAAUUACCAGCAAA)dTdT-3' anti-sense 5'-(UUUGCUGGUAAUUGCGUGG)dTdT-3'] was used in stress granules and immunoprecipitation-qPCR, NSC34 neurite extension experiments. The mix of 1 μg siRNA in 300 μl optiMEM with 0.8 μl lipofectamine was left for 20 min at room t° and added to cells for 3 h, after that the solution was removed and the usual media was added to the well. As control the negative siRNA (1027310, Qiagen) was applied in the same concentration as YB-1 siRNA.

**Preparation of plasmids for expression in mammalian cells**. Plasmids harboring the full length Lin28a, YB-1, G3BP1, FUS, CSDE1, LARP6, HuR, TDP43 genes fused with GFP and/or RFP/GFP-MBD were obtained previously[27] (see Table 1). To obtain the construct with N-terminal GFP fusion the vector pEGFP-C1 was used, for fusion with C-terminal RFP/GFP-MBD – vector pEF-DEST51. The same methodology was applied to prepare the plasmids carrying Lin28 truncated forms (N-ter 1–136 aa., C-ter 117–209 aa. in the text) fused with RFP-MBD.

Plasmids containing full length Lin28 mutated gene were obtained by the site-directed mutagenesis on the human *Lin28* gene directly on the Lin28-GFP-MBD and GFP–Lin28 plasmids. The mutagenesis experiments were performed by using the Quikchange II XL site-directed mutagenesis kit (Agilent) and the corresponding primers (Eurofins Genomics). DNA sequencing was used for verification of obtained plasmids.

**Microtubule bench assays**. HeLa cells were transiently transfected with the indicated plasmids for 24 h. Cells were washed with PBS, then fixed first with ice-cold methanol for 10 min at −20 °C and washed with PBS, then with PFA 4% for 30 min at 37 °C. The double fixation aims to improve the quality of microtubules final image. The RNA in situ hybridization was performed when necessary. The images of samples were acquired with Nikon Eclipse T*i* fluorescent microscope using the oil immersed 63×/1.4 NA objective.

To analyze of RBP interactions with the microtubule bench (one of the studied proteins is fused to GFP/RFP – MBD), RBPs colocalization on microtubules was meausred at single cell level using the method described previously[27]. To quantify the colocalization level between a protein bait fused to MBD and putative protein preys, we adapted a method previously described[61]. Both images were then filtered using a FFT high pass filter to remove spatial frequencies which are not relevant to microtubule structures (larger structures than 5 μm). Images of the bait and the prey were then merged into a single green-red image. Then, the ImageJ's plug-in, "PSC Colocalization", was used to measure the Spearman's coefficient, in three different regions of interest (ROI) for the same cell where microtubules are clearly

**Table 1 Plasmids used for expression in mammalian cells.**

| Fusion | RBP | RBP accession number | Figures where mentioned |
|---|---|---|---|
| GFP | Lin28 full length (wt/mut) | NP_078950.1 | 1a–b, 6a–c, S1c, S2b–c, S6a–d, S7a–b |
| | YB-1 | NP_004550.2 | 1a–b, S1c, S2b, S7a |
| | LARP6 | NP_060827.2 | 6a, S2b, S6a–b, S7a,b |
| | G3BP1 | NP_005745.1 | 1a–b, 5a, S1c, S2b, S7a |
| | TDP43 | NP_031401.1 | S7a |
| | CSDE1 | NP_001123995.1 | 1a–b, S1c |
| | HuR | NP_001410 | S7a |
| GFP-MBD | Lin28 full length (wt/mut) | NP_078950.1 | 4a–b, S2a, S4a |
| | YB-1 | NP_004550.2 | 4a–b |
| | G3BP1 | NP_005745.1 | 4a |
| RFP-MBD | Lin28 full length | NP_078950.1 | 1b–c, S1c |
| | Lin28 – N-ter (truncation, 1–136 aa.) | | S2b |
| | Lin28 – C-ter (truncation, 117–209 aa.) | | S2b |
| | YB-1 | NP_004550.2 | 1a–c, 4, S2c, S4a |
| | G3BP1 | NP_005745.1 | 1b–c, S1c |
| | FUS | NP_004951.1 | 1b–c, S1c |
| | HuR | NP_001410 | 1c |
| HA-tag | YB-1 | NP_004550.2 | 6b |

observed in the bait image. The area of the ROI was fixed to avoid any bias due to the surface considered to measure the correlation coefficient. (Fig. 1b, Supplementary Fig. S1c, Supplementary Data 1). The Spearman's coefficient is a better choice than the closely-related Pearson coefficient as it includes nonlinear relationship. Fluorescence intensity may increase non linearly with the number of baits or preys, especially when short-ranged non radiative interactions take place on microtubules at elevated bait or prey surface densities.

To measure sub-compartmentalization detection in the systems with both RBPs fused to GFP/RFP-MBD by image analysis, the cell image analysis was carried out as previously described[28]. Fluorescence analysis included processing of signal by filtering out large and small (shading and smoothing corrections) structures (Fast Fourier Transform process, FFT Bandpass filter tool, ImageJ) and removing the background intensity (Subtract background tool, ImageJ). The distribution of green and red fluorescence along microtubules was analyzed with a drawn line (thickness 4 pixels/100 nm, Freehand tool ImageJ, Supplementary Data 3). The analyzed length of the microtubule network was around 10 mm in total for each condition. A compartment was detected whenever fluctuation of the RFP/GFP fluorescence ratio exceeds 20%. The enrichment of the compartment was obtained by measuring the maximal ratio ($I_{RFP-YB-1}/I_{GFP-Lin28}$) or ($I_{GFP-Lin28}/I_{RFP-YB-1}$) over the length, L, of the considered compartment. To determine the larger contribution to RFP-YB-1 compartmenting, we consider the following Boolean tests: Log ($I_{RFP-YB-1}/$ mean ($I_{RFP-YB-1}$)) – Log ($I_{GFP-Lin28}/$ mean($I_{GFP-Lin28}$)) > 0 where the $I_{RFP-YB-1}/I_{GFP-Lin28}$ ratio is maximum. When the Boolean test gives True, RFP-YB-1 enrichment is considered as the major cause of compartmenting. When the Boolean test gives False, relative RFP-YB-1 enrichment is mostly due to the absence of GFP-Lin28. An analogous procedure was followed for analyzing GFP-Lin28-enriched compartments.

To measure the interactions of Lin28 truncated forms (fused to RFP-MBD) with YB-1-GFP, cross-sections of cell images were used. Then we analyzed their red and green channel profiles with ImageJ. Data was plotted as "prey" intensity (GFP-RBP) on microtubules versus "bait" intensity (RBP-RFP-MBD) on microtubules. The linear least squares fitting line is represented with the corresponding slope (Supplementary Fig. S2b).

*RNA-binding ability of Lin28 (wt and mutants fused to GFP-MBD).* Microtubules clusters were detected by CellProfiler software in green channel. For the analysis of protein-mRNA colocalization the cluster corresponding to microtubules and the cytoplasm around it were included in the studied area. The relative enrichment of mRNA on microtubules (red channel) was calculated as well as the same parameter for GFP signal in given area, then plotted as mRNA versus GFP enrichment on microtubules. Fitting was performed with a straight line (linear least squares, Supplementary Fig. S2a).

**Measure of neurite extensions.** To estimate neurite number and total neurite output NSC34 cells were fixed with paraformaldehyde (4%, w/v) after differentiation and at 48 h of transfection with corresponding plasmid and stained with the anti-α-tubulin E7 mouse primary antibody and goat anti-mouse secondary antibody (Invitrogen) to identify neuronal cells. Cell images were obtained using Carl Zeiss Axiovert 200 M fluorescent microscope. Neurites were manually traced using ImageJ software (version 1.46r, NIH) by tubuline channel. For each experiment, performed in triplicates, the data plotted represents the average of at

least 15 neurons expressing indicated GFP-labeled protein or indicated siRNA treatment (Supplementary Fig. S7a). The experiments have been performed in triplicate.

**In situ hybridization.** To visualize mRNA in red color, after fixation HeLa cells were incubated with oligo-dT-[Cy3], diluted in SSC 2X, 1 mg/ml yeast tRNA, 0.005% BSA, 10% dextran sulfate, 25% formamide, for 2 h at 37 °C. Wash steps were performed using 4X and then 2X SSC buffer (0.88% sodium citrate, 1.75% NaCl, pH 7.0). To visualize mRNA in blue color for SGs experiments, the oligo-dT with digoxigenin was used after cells fixation with the same incubation procedure as oligo-dT-Cy3. Then the primary anti-digoxigenin antibodies (mouse, ab420, Abcam) and secondary antibodies (goat anti-mouse, Alexa 350, Invitrogen) were applied to cells according to supplier's protocol.

**5-Bromodeoxyuridine (BrdU) incorporation analysis.** Hella cells, 72 h after transfection, were pulsed with 60 μM BrdU (Invitrogen) for 6 h at 37 °C. Cells were fixed with ice-cold methanol for 15 min at −20 °C, washed with PBS, after that fixed with paraformaldehyde (PAF) 4% for 25 min at 37 °C. After 3 wash steps with PBS, the cells were permeabilized with Triton X-100 0.3% during 15 min at room t°. Denaturation was performed with HCl 2 M for 30 min at 37 °C, then it was neutralized with Tris 0.1 M, pH 7.8, twice for 10 min. After 3 wash steps with PBS, Tween20 for 5 min, cells were kept with blocking buffer (PBS, 0.5% Tween20, 2% FBS) during 30 min at 37 °C. The primary anti-BrdU monoclonal rat antibodies (ab6326, Abcam) were diluted 1:1000 in blocking buffer and applied to cells for incubation overnight at 4 °C. After PBS washings, the secondary goat anti-rat antibody (Alexa 594, Invitrogen) were diluted 1:1000 in blocking buffer and added to cells for 1h30 at room t°. After PBS washing, staining with Fluoromount-DAPI (Sigma), diluted 1:8000 in PBS, was performed during 30 s followed with 3 washes PBS for 10 min each. The images were taken with Carl Zeiss Axiovert 200 M fluorescent microscope (Supplementary Fig. S7b).

BrdU incorporation was measured at single cell level using CellProfiler software, DAPI signal was used for nuclei detection. The signal intensity of overexpressed proteins fused with GFP was measured in cytoplasm, then data was plotted as distribution of BrdU-positive cells versus GFP integrated intensity (Supplementary Fig. S7b).

**Stress granules (SG) assay.** HeLa cells, transfected with corresponding plasmids for 24 h, were subjected to oxidative stress using 300 μM arsenite during 1 h at 37 °C. The cells were fixed with methanol for 20 min at −20 °C, followed with 4% PAF for 30 min at 37 °C. The staining was performed using anti-HA (against overexpressed YB-1, mouse, sc-7392, Santa Cruz Biotechnology) or anti-YB-1 (against endogenous protein, rabbit polyclonal, Bethyl Laboratories, Montgomery, USA) primary antibodies and then secondary antibody (goat anti-mouse/donkey anti-rabbit, Alexa 594, Invitrogen). RNA in situ hybridization was performed in some experiments. Carl Zeiss Axiovert 200 M fluorescent microscope was used to obtain the cell images. The SGs in cells were detected automatically using Cell-Profiler software. The overall cytoplasmic expression of proteins was measured as well as their signal intensity in SGs, their ratio gives an enrichment in SGs. The data was plotted as YB-1 versus GFP enrichment in SGs, the fitting (linear least squares) was performed and the slopes are present on the figures (Fig. 6c, Supplementary Fig. S6b, Supplementary Data 4).

**Proximity ligation assay (PLA).** For the proximity ligation assay (PLA) the kit from O-link Bioscience (Sweden) was used according to manufacturer's protocol. HeLa cells were grown and transfected with corresponding plasmids for 24 h. The cells were washed with PBS for 5 min, then fixed with 4% PAF for 20 min at 37 °C and washed in PBS. The cells were blocked using blocking solution (PBS, 3% BSA, 1% Triton) for 60 min at 37 °C. The anti-YB-1 and anti-GFP primary antibodies, diluted in blocking solution, were added to cells overnight at 4 °C, the samples were washed twice with PBS, 0.2% BSA, 0.1% Triton. The PLUS and MINUS PLA probes were diluted 1:5 in corresponding buffers, provided by manufacturer, and incubated with cells for 60 min at 37 °C, and the samples were washed twice for 5 min with 10 mM Tris, 150 mM NaCl, 0.05% Tween20. The ligase was diluted in ligation buffer 1:40 and applied to the cells for 30 min at 37 °C, then washed twice with 10 mM Tris, 150 mM NaCl, 0.05% Tween20. The solution of polymerase in amplification buffer 1:80 was added to the samples and incubated for 100 min at 37 °C. After that, the samples were washed with 200 mM Tris, 100 mM NaCl twice and mounted with Duolink In Situ Mounting Medium, containing DAPI, for 15 min. The cell images were obtained with Carl Zeiss Axiovert 200 M fluorescent microscope. PLA intensity was measured by probe (included in reaction buffer) fluorescence using CellProfiler software and plotted on graphs versus GFP intensity, measured in cell cytoplasm (Fig. 1d, Supplementary Fig. S6a, Supplementary Data 1).

**Immunoprecipitation.** HEK293 cells, grown and transfected for 48 h with indicated plasmids, were washed with PBS and lysed with a cell extraction buffer (25 mM Tris-HCl, 150 mM NaCl, 1% NP-40, 0.1% SDS, 0.5 mM DTT, 0.5U RNase inhibitor) for 1 h at 4 °C, then centrifuged at $23,000 \times g$ for 1 h at 4 °C. The supernatant was used for IP experiment.

Immunoprecipitation procedure was indicated by the supplier (Dynabeads kit (Thermofisher)). Firstly, for the preparation of beads they were resuspended in the vial by vortexing for 30 s, transferred to a tube and put on the magnet to separate them from the solution, then the supernatant was removed and the tube was taken away from magnet. Secondly, 5 μg of anti-GFP antibody (mouse, Merck) was diluted in PBS, Tween 20, and added to the beads followed with incubation at 4 °C for 1 h. Then the tube was placed on the magnet, and supernatant was removed.

Thirdly, the lysate of cells was added to the beads and gently resuspended with a pipette. The mix was incubated at 4 °C overnight. Then, the supernatant was then removed, and the beads were washed with the cell extraction buffer two times.

For RNA extraction, Trizol reagent was added to the bead fraction and the whole volume was gently mixed with a pipette several times, then left for 5 min at room temperature. After it was exposed to the magnet, the supernatant was finally taken.

**Extraction of RNA.** Chloroform (0.2 V) was added to the IP fraction and mixed for 15 s by tilding, then the sample was centrifuged at $12,000 \times g$ for 15 min at 4 °C. Supernatant was mixed with isopropanol (1 V) by tilding 6–7 times and left for 10 min at room t°. After centrifugation at $12,000 \times g$ for 15 min at 4 °C, the isopropanol fraction was removed by pipetting and evaporation, 70% EtOH (1 V) was added to the pellet, and the suspension was centrifuged at $12,000 \times g$ for 7 min at 4 °C. The supernatant was removed, and the pellet was left to dry out for 5 min at room t°, then it was dissolved in ultrapure $H_2O$ on ice for 10 min with gentle shaking and at 70 °C for 20 min with vortexing. The next step was phenol/chloroform extraction. The phenol + chloroform mix (1 V) was added to the solution and vortexed for 3 s, then centrifuged at $12000 \times g$ for 10 min at 12 °C. The supernatant was transferred to another tube and mixed with chloroform (0.46 V), then centrifuged at $12,000 \times g$ for 10 min at 12 °C. To the obtained supernatant 2 M NaAc, pH 5.2 0.1 V, 96% EtOH 2 V was added, left for 30 min at −20 °C, then centrifuged at $12,000 \times g$ for 30 min at 4 °C. The pellet was washed with 70% EtOH, centrifuged again at $12,000 \times g$ for 15 min, dried out and diluted in water for 20 min at 70 °C, then the RNA concentration in the sample was measured using Nanodrop One (ThermoFisher).

**RT-qPCR.** For the reverse transcription reaction (RT) the mix of random primers 5 μM, RNA 1 μg, nuclease-free $H_2O$ was prepared on ice, then kept for 5 min at 65 °C and for 5 min on ice. After that, dNTPs till 0.5 mM, reaction buffer, 3 mM $MgCl_2$, RNase inhibitor, $H_2O$ and reverse transcriptase ProtoScript (Biolabs) were added to the previous mix, placed for 10 min at 25 °C, then 50 min at 42 °C for reaction and 15 min at 70 °C for the enzyme inactivation.

For qPCR (Fig. 6d) the kit (Promega) was used according to the standard procedure. Primers for the genes (*oaz1, act, ctnnb1, b2m, eif4g1, eif5a, gapdh, rpl8, eef1a1, eef2, ddx17, pkm2*) were created with Primer-BLAST online service (NCBI) (see Table 2). Each reaction mix contained forward and reverse primers 1 μl each till 5 μM, cDNA 8 μl, $H_2O$ 1 μl and 10 μl of the qPCR mix from the kit. The 96-well plates were used, the amplification was performed on Applied Biosystems 7500 Real Time PCR System. Data obtained with the primers presented in Table 2 are shown in Supplementary Data 4.

**Western blot and IP analysis.** SDS-PAGE was performed according to the standard protocol using a prestained molecular weight marker. Transfer of SDS-PAGE to a membrane was realized by electric current application at 80 V for 1.5–2 h on ice in a Tris 25 mM, glycine 192 mM buffer with EtOH 20%. After the

## Table 2  Primers used in RT-qPCR.

| Gene | Forward primer | Reverse primer |
|---|---|---|
| OAZ1 | TACAGCAGTGGAGGGAGACC | GGATAAACCCAGCGCCAC |
| ACT | CATGTACGTTGCTATCCAGGC | CTCCTTAATGTCACGCACGAT |
| CTNNB1 | AAAGCGGCTGTTAGTCACTGG | CGAGTCATTGCATACTGTCCAT |
| B2M | TCTCTGCTGGATGACGTGAG | TAGCTGTGCTCGCGCTACT |
| EIF4G1 | CCCAACTGTAGAAGGCATCC | CTCCAGGCCCTTGTAGTGAC |
| EIF5A | CATTGGGAAGGTGGCTGA | GGGTCGAGTCAGTGCGTT |
| GAPDH | CCTCCTGCACCACCAACTGCTTA | GTGATGGCATGGACTGTGGTCAT |
| RPL8 | AGATGGGTTTGTCAATTCGG | CAAGAAGACCCGTGTGAAGC |
| EEF1A1 | ACTTGCCCGAATCTACGTGT | TTGCCGCCAGAACACAG |
| EEF2 | GCACGTTCGACTCTTCACTG | CTGGAGATCTGCCTGAAGGA |
| DDX17 | TCCATCATGCTAACTTCCCACA | CGGAAATCCCTGGCACTGAA |
| PKM2 | GTCTGAATGAAGGCAGTCCC | TCCGGATCTCTTCGTCTTTG |

transfer, the membrane was stained with ponceau S red 0,2% solution to control the transfer and detect the total protein. The stained membrane was agitated with $CH_3COOH$ 1% for fixation, then washed with TBS-Tween buffer (20 mM Trizma Base, 143 mM NaCl, pH 7.6, 1% Tween20) and blocked with non-fat dry milk 5% for 40 min shaking at room t°. Then, the membrane was washed again with TBS-Tween and agitated with anti-YB-1 primary antibodies (Abcam, n 12148, rabbit, 1:1000) in 5% milk shaking overnight at 8 °C. After a wash step, the secondary antibody (LI-COR IRDye, IR-Long 800CW, goat anti-rabbit, 1:4000) was added to the membrane in 2% milk for 45 min at room t° shaking. Then, the membrane was washed and scanned with Amersham Typhoon bioimager to detect the bands corresponding to YB-1. After the first scan, the membrane was agitated with anti-GFP primary antibodies (SantaCruz Biotechnologies, SC8334, rabbit, 1:1000) for 1.5–2 h at room temperature. Then, the same steps were repeated, and the membrane was scanned again to detect the GFP and YB-1 bands (Fig. 6a).

**Protein expression data across tissues and during embryogenesis.** To find any RBPs expression correlation in different tissues (Fig. 7a, b), the RNA-seq data was collected from the Human Protein Atlas database in category "RNA binding" and filter for "RNA human"[46]. All found 48 RBPs are abundant in all tissues, they were ranked according to their expression score (sum of mRNA-seq values over all tissues) (Supplementary Data 6). Lin28 is known to be expressed in limited amount of tissues, so its scores were added to the table. The correlation was measured by using the Pearson coefficient.

Analysis of RBP RNA expression during embryogenesis was performed on the data obtained at the single cell level[47] and plotted as reads per kilo-mapped-reads (RPKM) versus the step of embryonic development (Fig. 7)

**Statistics and reproducibility.** Statistical tests, sample size, and number of biological replicates are reported in all the figure legends and/or described in the method sections. Student's or Kolmogorov–Smirnov's tests were used to compare all the mutants with each other. For the microtubule bench experiments presented in Fig. 4. the set of data containing the number of clusters, their size and enrichment was obtained and compared with data set for control experiments (wild type Lin28) using *t*-test with two tails and Kolmogorov–Smirnov test (ks-test) to detect the significant difference between taken populations. *P* values were calculated for every tested pair mutant – wt, and 4 passive mutations were used after as controls since their distribution is not significantly different from each other and wild type Lin28. To report the highly significant mutations, which were selected for cell experiments, the ks-test was used as more relevant and reliable, however, these chosen mutations were also showing the significant difference from controls using *t*-test (Supplementary Fig. S4b).

**Reporting summary.** Further information on research design is available in the Nature Research Reporting Summary linked to this article.

## Data availability

Source data underlying the graphs presented in the main figures are available in the Supplementary Data files. Inquiry of any additional data should be requested to the corresponding author.

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

## Acknowledgements

The authors would like to thank INSERM (PRI, RaPiD) and University of Evry for continuous support of the SABNP laboratory. This study was also supported by MSD France (PhD grant to A.S.), Genopole (Grant for postdoctoral fellowship support to K.E.H.), the Doctoral School SDSV of University Paris-Saclay for a course grant (to A.S.), Région Ile-de-France (SESAME) [15013102, in part] and the EU (Marie Skłodowska-Curie Individual Fellowships Grant 'MITiC' (895024) to K.E.H.) Funding for open access charge: INSERM. We also grateful to Dmitry Kretov for plasmid preparation and, Piotr Sliz and Lonfei Wang for providing their NMR data about Lin28 (HSQC data) that was very useful.

## Author contributions

A.S. performed the NMR experiments and analyzed the data. K.E.H. designed and performed the MD simulation and the analysis of the interactions. V.J. and A.S. prepared the expression plasmids used in this study. B.D. performed the MT bench assays and

analyzed the data. B.D. performed the proliferation assays, stress granule experiments and the analysis of neurite extensions. M.-J.C. and E.S. supervised the NMR experiments and discussed the results with A.S. A.S., G.L. and A.B. produced the proteins. H.H., N.B. and P.C. participate in the development of the MT bench assay. R.C.M., A.M., A.B., E.S., D.L., L.O., L.H. discussed the results, commented on the manuscript, and contributed to its final version. D.P conceived the project and wrote the manuscript.

## Competing interests

The authors declare no competing interests.
