## [Peer Review File · Communications Biology]

Reviewers' comments:

Reviewer #1 (Remarks to the Author):

This manuscript from Samsonova et al explores the relationship between YB-1 and Lin28 binding in vitro and in vivo. Although combining an “intracellular bench” and structural studies is an innovative approach to look at this issue, the paper overstates its conclusions. At several points, alternative explanations are not considered, and there is often a lack of positive and negative controls. Moreover, to support the conclusions made by the authors, orthogonal techniques (such as immunoprecipitation) need to be used, and the authors need to directly test their hypothesis and model in terms of RNA binding and impact on gene expression. Finally, the current CLIP analysis is incomplete, lacks in statistical tests and analyses, and is very difficult to follow. In summary, although these results have potential to be interesting, the current version of manuscript has fundamental flaws that must be addressed before I can support publication.

Minor comments:

- 1) The language and writing (especially grammar and syntax) are very difficult to follow.
- 2) Consider accessibility in choice of colors: the current figures will not be accessible to those with red-green color blindness.

Reviewer #2 (Remarks to the Author):

Rather than focusing on the function of Lin28 in processing the miRNA, let-7, Samsonova et al. study the role of Lin28 protein in co-localizing with RBP YB-1, suggesting an unappreciated role for Lin28 in RNA regulation. The authors show that Lin28 and YB-1 co-localize in vivo (via a microtubule tethering assay, and more convincingly, by PAR and stress granule localization). There is an interesting correlation that proteins with CSD (cold shock domain) can co-localize, while other RNA binding proteins are excluded. They used NMR of Lin28 and YB-1 CSD with RNA to identify residues that are likely important in allowing both proteins to bind the same single stranded DNA (i.e. the cooperative function referred to in the text). They test these residues in the microtubule tethering assay and find mutations in key residues do limit mixing of Lin28 and YB-1 and model Lin28 / YB-1 complexes using MD to show that they could interact with RNA via their CSDs. The microtubule tethering assay localizes the proteins to a non-physiological location (the benefit of this assay and the localization change is not made clear in the paper). However, they also look at the localization in stress granules, which is more physiological, and find that forcing expression of Lin28 in cells will drive YB-1 into stress granules. When YB-1 levels are lowered by knockdown, YB-1 enrichment in stress granules increases in cells with WT Lin28, but not with the lin28-RS mutant.

The paper offers evidence that Lin28 and YB-1 affect one another's co-localization within cells and identifies residues in Lin28 that are important for that co-localization, exploring another function for Lin28 outside of maturation of let-7 miRNA. However, the paper makes claims about whether co-localization via microscopy equals direct protein-protein interaction and makes claims about the functional consequences of this interaction between Lin28 and YB-1 on translation regulation that are not warranted by the data. There are also several references to how the two proteins pack onto RNA and exclude other RNPs, which are also not supported through experimentation.

In its current state, the paper could give a reader the impression that the functional consequence and physiological relevance of Lin28/YB-1 co-localization is clarified by this work when it is not. This overreach of conclusions starts in the abstract with “Lin28 thus occupies a unique position to efficiently control the expression of many mRNPs by using YB-1 as an “entry badge” whereas most RNA-binding proteins, with the exception of few proteins such as CSDE1 and Lin28b, are denied access for lack of CSD.” While other examples of overreaching conclusions are listed below, this list is not exhaustive. While the exploration of co-localization between Lin28 and YB-1 will interested readers, there is too many unsubstantiated claims in the paper’s current form. Either the conclusions should be more strictly based on the data shown in the paper (and speculation on how this might inform function in translation regulation should be clearly presented as models and hypotheses for future work) or additional experiments need to be added to support the models on how Lin28/YB-1 co-localization is functionally important. I think the paper would still be interesting to the Communications Biology audience with the current data (with the addition of a co-IP between Lin28 and YB-1, with RNase treatment), but written more carefully to avoid overinterpretation of the functional consequences of the Lin28 and YB-1 co-localization (or perhaps, direct interaction).

Major Points:

- Throughout the paper, the authors refer to interactions between Lin28 and YB-1. Many readers would interpret this as direct protein-protein interactions. While the authors refer to co-IPs in their previous work (Boca et al.) as evidence for a direct interaction between Lin28 and YB-1, these IPs were not completed in the presence of RNases. So, the observed pulldown between Lin28 and YB-1 could be mediated by RNA. The authors should repeat this immunoprecipitation +/- RNases to clarify whether the interaction between Lin28 and YB-1 is mediated by RNA to clarify how direct the protein-protein interaction is. The microtubule tethering assay that they rely on in this paper cannot differentiate between the two. What’s more, it is not clear that the microtubule tethering assay has the resolution to determine whether the proteins are in close proximity or on the same RNA. Therefore, without more direct biochemical evidence of a direct interaction, the term “co-localization” would be better suited to this data than “interaction”
- While the data shows that Lin28 and YB-1 can co-localize with the same microtubules, I wonder about the physiological relevance of such an assay. It is not clear why using microtubules as a nanoplatform is beneficial for probing the interaction between Lin28 and YB-1. It is likely that tethering each protein to a microtubule would encourage interactions that would normally be weaker in the cytoplasm.
- The PAR data and the microtubule nanoplatform assays can differ show different trends with the same mutant, as the authors state: “In addition, using proximity ligation assays, we found that Lin28-QS, -RS and -DE still colocalize with endogenous YB-1 in cells (Supplementary Fig. S6a). However, when YB-1 was used as bait to bring Lin28 on microtubules, Lin28-QS and -RS mutants formed distinct compartments on microtubules” This seems to indicate that the microtubule assay isn’t testing a physiological interaction. Or perhaps the authors have some reason to think that this interaction is better able to detect small perturbances? Figure 5b: How can the authors determine “This behavior reflects the fact that the more the granules recruit mRNA the more the enrichment of RNA-binding proteins increases” when they were tracking YB-1 and Lin28, but not mRNA.
- Fig. 5c: the rest of the data in the paper suggests that Lin28 and YB-1 cooperate in binding to nucleic acid. Here the Lin28-RS mutant shows decreased association with mRNA, perhaps because of weaker binding to YB-1 (as the RS mutant shows decreased localization with YB-1). Yet, knockdown of YB-1 restores Lin28-RS binding to mRNAs. This seems to contradict the model that Lin28 and YB-1 cooperate. The authors should expand on why this data contradicts the rest of their findings and what it could mean.

- Authors say “The results provided here demonstrate that Lin28, independently of the repression of let-7 maturation, could reprogram global mRNA translation through a cooperative association with YB-1 to mRNPs relying on the structural similarity of their CSD”; I don’t see the data to support that statement. They need to show that the “active” Lin28 mutations shown in Fig. 3b interact more weakly with YB-1 via co-IP (in the presence of RNases) and that these Lin28 mutants alter the translation rate a large number of mRNAs. Change in co-IP affinities of a few mRNAs with Lin28-RS does not mean that this mutant is sufficient to alter their translation.
- This statement is not supported by the data in the paper: “All the other RBPs (G32BP-1, FUS, TDP-43, LARP6, HuR were tested here) having different RNA-binding domains cannot fit into the YB-1-rich mRNPs due to the tight packing of CSDs along mRNA (Figure 2c and 4a-b).” Fig. 2c shows that both Lin28 and YB-1 can bind single stranded nucleic acid. But whether they form a complex on the same nucleic acid is not clear, as there is no convincing size shift to reflect both proteins binding the same piece of nucleic acid. Perhaps electrophoresis conditions with more resolving power is needed? And there is no experiment to test whether these other RBPs are blocked by binding of Lin28 and YB-1. While the MD experiments in Fig. 4 model that a trimer of Lin28-RRM TDP-43- Lin28 would have weaker binding than a trimer of Lin28 - YB-1 - Lin28, neither trimer is known to be physiologically relevant.
- Likewise, there is no experimental support for the statement “Lin28 thus occupies an interesting position in the RNA-binding protein kingdom to control the activation/inactivation of dormant mRNPs that are mostly packaged by YB-1.”, as there is no evidence presented in the data that Lin28 can control activation/inactivation of any mRNA, much less those that are specifically packaged by YB-1.
- The analysis of data sets from other papers on Lin28 and YB-1 CLIP and ribosome occupancy are not convincing, as they are from different experiments and different cell lines. This should be removed or the key points from the data exploration should at least be validated for some RNAs with the most significant trends.

Minor Points:

- Are all of the prey protein/GFP fusions (fig. 1) ever expressed by themselves to ensure that they don’t interact with microtubules on their own (or with RNPs that are known to interact with microtubules)? Please add that control data to the supplementary figures or include references to those controls if they are all previously published.
- Fig. 1a legend: need to specify what that top and bottom grayscale images are (top = bait; bottom = prey?)
- Fig. 1c: I’m very surprised that this clear pattern of mixing and de-mixing works in this microtubule tethering assay. I would have assumed that both bait and prey could bind microtubules randomly and either create an overall mixed signal (yellow), if low resolution, or puncta along microtubules of red and green. However, the bait and prey (for example in YB-1 and HuR) are localizing to mutually exclusive microtubules rather than exclusive portions of the same microtubules. Do the authors have insight into why that might be the case? Again, if the bait and prey were expressed alone, wouldn’t each be expected to bind microtubules non-specifically and show similar patterns?
- Fig 3c: legend should indicate what the color coding means
- Figure 5a: LARP6/YB-1 merged image looks yellow, suggesting overlap, but the quantitation doesn’t reflect that. Why does it look like YB-1 is in these stress granules? Showing the red and green channels separately below the merged images would be clearer than showing red/green merged and mRNA channels separately. Then the reader could see if YB-1 is concentrated in the red channel in the same place as the stress granule.

Response to Reviewer's

Reviewer #1 (Remarks to the Author):

“This manuscript from Samsonova et al explores the relationship between YB-1 and Lin28 binding in vitro and in vivo. Although combining an “intracellular bench” and structural studies is an innovative approach to look at this issue, the paper overstates its conclusions. At several points, alternative explanations are not considered, and there is often a lack of positive and negative controls. Moreover, to support the conclusions made by the authors, orthogonal techniques (such as immunoprecipitation) need to be used, “We agree with the reviewer that the use of orthogonal techniques (immunoprecipitation, IP) should be used to challenge the results obtained with microtubule bench assay or by measuring the enrichment of proteins in stress granules.

We re-performed extensive analyses of the interaction of Lin28 with YB-1 (Fig. 6a). The results indicate clearly that YB-1 co-precipitate with Lin28 in the presence of RNA but not after RNase treatment. This is in agreement with our model of a cooperative binding of Lin28 and YB-1 to mRNA through their cold-shock domain.

We also probed the co-sedimentation of YB-1 with wild-type and Lin28 mutants. While we still detected Lin28 mutants in the IP fraction, the co-precipitation efficiency is reduced compared to wild type Lin28 (triplicates are shown in Fig. 6a).

We also provide additional controls for the microtubule bench assays (Fig. S2d).

“and the authors need to directly test their hypothesis and model in terms of RNA binding and impact on gene expression. Finally, the current CLIP analysis is incomplete, lacks in statistical tests and analyses, and is very difficult to follow. In summary, although these results have potential to be interesting, the current version of manuscript has fundamental flaws that must be addressed before I can support publication. “

We tested the binding of Lin28 to mRNA and found it to be dependent on YB-1 expression (IP and RT-PCR analyses). However, we agree with the reviewer that we did not investigate the functional consequences in terms of translation efficiency. We previously used CLIP data analysis for that purpose because we cannot draw conclusions based on the analysis of few mRNAs. Genome wide data are better suited for this purpose. However, Reviewer 2 considers that CLIP data should not be presented because we compared genome wide data from different cell lines (Lin28 CLIP from mouse stem cells (endogenous expression) and YB-1 CLIP data from human glioblastoma cells). Reviewer 2 also suggests to remove this part that may be explored in a separate paper. It makes sense owing to the amount of data to be provided (CLIP analysis of endogenous Lin28 and silencing YB-1 in the same cell lines with analysis of the presence of mRNAs in polysome or non polysomal fraction at the genome wide level). As reviewer 1 raised also issue about the CLIP data, we therefore agreed with reviewer 2 to withdraw this part of the results and to concentrate on the structural basis on the interaction between Lin28 and YB-1 with validation in a cellular context. We just emphasize in the revised version the intriguing co-

expression pattern of Lin28 and YB-1 in human tissues compared to other RNA-binding proteins and similar temporal expression of Lin28 and YB-1 during embryo development.

Minor comments:

1) The language and writing (especially grammar and syntax) are very difficult to follow.

OK. We tried to do our best to correct the grammatical and syntax errors.

2) Consider accessibility in choice of colors: the current figures will not be accessible to those with red-green color blindness.

We have now provided images in different colors, provide higher magnifications images with separated colors, when necessary, or measurements to allow the accessibility to those with red-green color blindness.

Reviewer #2

Rather than focusing on the function of Lin28 in processing the miRNA, let-7, Samsonova et al. study the role of Lin28 protein in co-localizing with RBP YB-1, suggesting an unappreciated role for Lin28 in RNA regulation. The authors show that Lin28 and YB-1 co-localize in vivo (via a microtubule tethering assay, and more convincingly, by PAR and stress granule localization). There is an interesting correlation that proteins with CSD (cold shock domain) can co-localize, while other RNA binding proteins are excluded. They used NMR of Lin28 and YB-1 CSD with RNA to identify residues that are likely important in allowing both proteins to bind the same single stranded DNA (i.e. the cooperative function referred to in the text). They test these residues in the microtubule tethering assay and find mutations in key residues do limit mixing of Lin28 and YB-1 and model Lin28 / YB-1 complexes using MD to show that they could interact with RNA via their CSDs.

The microtubule tethering assay localizes the proteins to a non-physiological location (the benefit of this assay and the localization change is not made clear in the paper). However, they also look at the localization in stress granules, which is more physiological, and find that forcing expression of Lin28 in cells will drive YB-1 into stress granules. When YB-1 levels are lowered by knockdown, YB-1 enrichment in stress granules increases in cells with WT Lin28, but not with the lin28-RS mutant.

The paper offers evidence that Lin28 and YB-1 affect one another's co-localization within cells and identifies residues in Lin28 that are important for that co-localization, exploring another function for Lin28 outside of maturation of let-7 miRNA. However, the paper makes claims about whether co-localization via microscopy equals direct protein-protein interaction and makes claims about the functional consequences of this interaction between Lin28 and YB-1 on translation regulation that are not warranted by the data. There are also several references to how the two proteins pack onto RNA and exclude other RNPs, which are also not supported through experimentation.

In its current state, the paper could give a reader the impression that the functional consequence and physiological relevance of Lin28/YB-1 co-localization is clarified by this work when it is not. This

overreach of conclusions starts in the abstract with “Lin28 thus occupies a unique position to efficiently control the expression of many mRNPs by using YB-1 as an “entry badge” whereas most RNA-binding proteins, with the exception of few proteins such as CSDE1 and Lin28b, are denied access for lack of CSD.” While other examples of overreaching conclusions are listed below, this list is not exhaustive. While the exploration of co-localization between Lin28 and YB-1 will interest readers, there is too many unsubstantiated claims in the paper’s current form. Either the conclusions should be more strictly based on the data shown in the paper (and speculation on how this might inform function in translation regulation should be clearly presented as models and hypotheses for future work) or additional experiments need to be added to support the models on how Lin28/YB-1 co-localization is functionally important.

I think the paper would still be interesting to the Communications Biology audience with the current data (with the addition of a co-IP between Lin28 and YB-1, with RNase treatment), but written more carefully to avoid overinterpretation of the functional consequences of the Lin28 and YB-1 co-localization (or perhaps, direct interaction).

We understand the concerns of the reviewers. We therefore rephrase the abstract and the discussion section to prevent an overreach of conclusions. According to the reviewers, the functional consequences of the co-localization between Lin28 and YB-1 in terms of translational regulation needs to be presented in a separated paper. In addition, it would require to gather a significant amount of data (CLIP analysis of polysomes and redistribution of mRNA between non polysomal and polysomal pools in cells expressing Lin28 endogenously, etc.).

We have added IP data as proposed by the reviewer.

Major

Points:

- Throughout the paper, the authors refer to interactions between Lin28 and YB-1. Many readers would interpret this as direct protein-protein interactions. While the authors refer to co-IPs in their previous work (Boca et al.) as evidence for a direct interaction between Lin28 and YB-1, these IPs were not completed in the presence of RNases. So, the observed pulldown between Lin28 and YB-1 could be mediated by RNA. The authors should repeat this immunoprecipitation +/- RNases to clarify whether the interaction between Lin28 and YB-1 is mediated by RNA to clarify how direct the protein-protein interaction is. The microtubule tethering assay that they rely on in this paper cannot differentiate between the two. What’s more, it is not clear that the microtubule tethering assay has the resolution to determine whether the proteins are in close proximity or on the same RNA. Therefore, without more direct biochemical evidence of a direct interaction, the term “co-localization” would be better suited to this data than “interaction”.

We have performed the immunoprecipitation experiments with and without RNases. The results indicate that the interaction between Lin28 and YB-1 is mediated by mRNA, as expected (Fig. 6a).

We agree with the reviewer 1 that the term “interaction” can be ambiguous. We then used the term “co-localization” for the results obtained with the MT bench assays. We also precise that the interaction is not direct and relies on the presence of mRNA (cooperative association without direct interaction).

- While the data shows that Lin28 and YB-1 can co-localize with the same microtubules, I wonder about the physiological relevance of such an assay. It is not clear why using microtubules as a nanoplatform is beneficial for probing the interaction between Lin28 and YB-1. It is likely that tethering each protein to a microtubule would encourage interactions that would normally be weaker in the cytoplasm.

We understand the concern of the reviewer. The point here is to confine the two RBPs on microtubules to promote the formation of mRNA-rich compartments (Maucuer et al., J Cell Science, 2018) (encouraging interactions, as mentioned by the reviewer). This allows to analyze whether the two compartments (RFP and GFP- labelled) are mixing or demixing. If the two proteins bind to the same RNA via a cooperative binding or have heterotypic interactions with each other, we may expect them to mix. This is what we observed with lin28 and YB-1. However, a co-localization was not found by other RNA-binding protein such as HuR, FUS, and G3BP-1 that do not have a cold-shock domain.

However, we agree with the reviewer that this system is “artificial”. PLA and, now in the revised version, immunoprecipitation assays confirm the co-localization between Lin28 and YB-1 that relies on the presence of mRNA (RNase treatment).

This MT bench assay is also very interesting because it is sensitive enough to detect weak perturbation in the mixing between two proteins and was therefore used to screen the effect of mutations on RBP mixing. For example, the F47A mutation (Lin28) impairs the binding of the Lin28 CSD to mRNA (conserved residue interacting directly with nucleic acids). With this mutant, we observed a striking demixing with YB-1, which is in agreement with a cooperative association of YB-1 and Lin28 to mRNA.

In the revised version, we also show that the identified mutations (RS, QS) affect the efficiency of the pull down of YB-1 by Lin28 (Fig. 6a).

- The PAR data and the microtubule nanoplatform assays can differ show different trends with the same mutant, as the authors state: “In addition, using proximity ligation assays, we found that Lin28-QS, -RS and -DE still colocalize with endogenous YB-1 in cells (Supplementary Fig. S6a). However, when YB-1 was used as bait to bring Lin28 on microtubules, Lin28-QS and -RS mutants formed distinct compartments on microtubules” This seems to indicate that the microtubule assay isn’t testing a physiological interaction. Or perhaps the authors have some reason to think that this interaction is better able to detect small perturbances?

This is an interesting remark. Proximity ligation assays (PLA) indeed indicate a co-localization between Lin28-RS and QS mutants with YB-1. When YB-1 was used as a bait to bring Lin28 on microtubules, we still observed that Lin28-RS and QS mutants were brought onto microtubules but in contrast with wild type Lin28, we observed a tendency to form separate compartment onto microtubules (Figure S2c). We think that the Lin28-RS and QS mutants still co-localize with YB-1 but their “interaction” is altered. It is also observed when both wild type YB-1 and Lin28-RS or QS mutants are confined on microtubules, as we observed partial demixing compared to wild type Lin28 (Figure 3B) but not a strong demixing (see YB-1 and G3BP-1, Fig. 4a). Proximity ligation assays may not be sensitive enough to detect an altered cooperative association to mRNA because large separation distance (up to 40 nm) can induce a PLA

signal. In the revised version, we added IP data indicating a still present but less marked co-localization of Lin28-RS and -QS mutant with YB-1 (Fig. 6a).

The text was rephrased to indicate this point.

Figure 5b: How can the authors determine “This behavior reflects the fact that the more the granules recruit mRNA the more the enrichment of RNA-binding proteins increases” when they were tracking YB-1 and Lin28, but not mRNA.

Maybe the sentence was misleading. We noticed a positive correlation between the enrichment of Lin28 and YB-1 but this is also observed with LARP6 and YB-1. We think that this correlation is consecutive to the fact that some stress granules may have more RBPs than others (not necessarily related to the size of stress granules but also the density of RBPs in granules). The sentence has been changed.

- Fig. 5c: the rest of the data in the paper suggests that Lin28 and YB-1 cooperate in binding to nucleic acid. Here the Lin28-RS mutant shows decreased association with mRNA, perhaps because of weaker binding to YB-1 (as the RS mutant shows decreased localization with YB-1). Yet, knockdown of YB-1 restores Lin28-RS binding to mRNAs. This seems to contradict the model that Lin28 and YB-1 cooperate. The authors should expand on why this data contradicts the rest of their findings and what it could mean.

We understand the concerns of reviewer 2. We now provide an explanation for this. When we compared the Lin28 binding to mRNA to that of Lin28-RS, we found that Lin28-RS is less efficient for binding cellular mRNA. We think that disrupting the cooperative association of Lin28 and YB-1 to mRNA prevents Lin28-RS to gain access to YB-1 rich mRNA. YB-1 therefore acts as a competitor for the binding of Lin28-RS to mRNA. Accordingly, decreasing YB-1 allows to increase the binding of Lin28-RS to mRNA in the absence of YB-1. The text was rephrased to emphasize this point. This pattern is not observed with wild type Lin28 because it has the ability to replace YB-1 in mRNPs but can also bind mRNAs on its own when YB-1 levels are decreased.

- Authors say “The results provided here demonstrate that Lin28, independently of the repression of let-7 maturation, could reprogram global mRNA translation through a cooperative association with YB-1 to mRNPs relying on the structural similarity of their CSD”; I don’t see the data to support that statement. They need to show that the “active” Lin28 mutations shown in Fig. 3b interact more weakly with YB-1 via co-IP (in the presence of RNases) and that these Lin28 mutants alter the translation rate a large number of mRNAs. Change in co-IP affinities of a few mRNAs with Lin28-RS does not mean that this mutant is sufficient to alter their translation.

We now provide Co-IP data with the Lin28 mutants that show their reduced efficiency to co-immunoprecipitate YB-1 (Fig. 6a). We also show that the presence of mRNA is required for their interaction as expected for a cooperative association to mRNA without direct interactions (Fig. 6a, RNase treatment).

We agree that affinities for mRNA do not mean changes in their translation rate. The sentence indicated by the reviewer has been rephrased. We clearly indicate that this is a perspective based on our data showing a cooperative association of YB-1 and Lin28 to mRNA.

- This statement is not supported by the data in the paper: “All the other RBPs (G32BP-1, FUS, TDP-43, LARP6, HuR were tested here) having different RNA-binding domains 44 cannot fit into the YB-1-rich mRNPs due to the tight packing of CSDs along mRNA (Figure 2c and 4a-b).”

This statement has been removed (“all RBPs”) and we only present this model as an hypothesis that is in agreement with the experimental results presented in this manuscript (MT bench assays (Fig.1b, Fig. 1c and Fig. 2C and new Fig S3a)

Fig. 2c shows that both Lin28 and YB-1 can bind single stranded nucleic acid. But whether they form a complex on the same nucleic acid is not clear, as there is no convincing size shift to reflect both proteins binding the same piece of nucleic acid. Perhaps electrophoresis conditions with more resolving power is needed?

In the revised manuscript, we present new gel shift assays for a larger range of YB-1 concentrations to show the decrease in the electrophoretic mobility of YB-1-rich mRNPs due to the presence of Lin28 (Fig. 2C). At lower concentration, the effect was also observed but less marked, as noticed by the reviewer.

And there is no experiment to test whether these other RBPs are blocked by binding of Lin28 and YB-1.

We now add an additional experiment with FUS, in complement to what we already done with TDP-43. Here we do not consider that YB-1 is blocking the TDP-43 or FUS binding of mRNA but rather that FUS and TDP-43 bind to different mRNAs leading to the presence of two distinct bands (Fig. S3a).

While the MD experiments in Fig. 4 model that a trimer of Lin28-RRM TDP-43- Lin28 would have weaker binding than a trimer of Lin28 -YB-1 - Lin28, neither trimer is known to be physiologically relevant.

Yes, we agree with the reviewer. The YB-1 multimer has been observed in vitro and validated by NMR analysis in our previous study (Kretov, NAR, 2019). This model provides a good model to study by MD the cooperative binding of the cold-shock domain to mRNA that may drive Lin28 and YB-1 co-localization in the presence of mRNA. It is now clearly indicated in the text.

- Likewise, there is no experimental support for the statement “Lin28 thus occupies an interesting position in the RNA-binding protein kingdom to control the activation/inactivation of dormant mRNPs that are mostly packaged by YB-1.”, as there is no evidence presented in the data that Lin28 can control activation/inactivation of any mRNA, much less those that are specifically packaged by YB-1.

The text has now been rephrased to present this statement as an interesting hypothesis for future investigations.

- The analysis of data sets from other papers on Lin28 and YB-1 CLIP and ribosome occupancy are not convincing, as they are from different experiments and different cell lines. This should be removed or the key points from the data exploration should at least be validated for some RNAs with the most significant trends.

We agree with the reviewer's remarks.

Minor

Points:

- Are all of the prey protein/GFP fusions (fig. 1) ever expressed by themselves to ensure that they don't interact with microtubules on their own (or with RNPs that are known to interact with microtubules)? Please add that control data to the supplementary figures or include references to those controls if they are all previously published.

We performed a new experiment to show that preys expressed alone do not localized on microtubules (no fusion with an MBD domain, Figure S2d). This control was indeed missing.

- Fig. 1a legend: need to specify what that top and bottom grayscale images are (top = bait; bottom = prey?)

OK. It is now better indicated in the figure.

- Fig. 1c: I'm very surprised that this clear pattern of mixing and de-mixing works in this microtubule tethering assay. I would have assumed that both bait and prey could bind microtubules randomly and either create an overall mixed signal (yellow), if low resolution, or puncta along microtubules of red and green. However, the bait and prey (for example in YB-1 and HuR) are localizing to mutually exclusive microtubules rather than exclusive portions of the same microtubules. Do the authors have insight into why that might be the case?

This demixing pattern may be due to a liquid-liquid phase separation mechanism. We already observed this pattern for TDP-43, FUS and HuR (Maucuer et al., J Cell Science, 2018). We think that YB-1 may exclude other RBPs by its cooperative association to mRNA, but also possibly with its long and unstructured self- adhesive CTD .

It is true that two different RBPs brought onto microtubules may appear as localized to mutually exclusive microtubules rather than to exclusive portions of the same microtubules. However, there are also puncta along microtubules of red and green. This pattern was also observed with TDP-43 (Maucuer et al., 2018, Fig. 3 and Fig.5) and is not specific to YB-1. Most probably, RBP-rich compartments can extend along a long distance on microtubules in this assay.

Again, if the bait and prey were expressed alone, wouldn't each be expected to bind microtubules non-specifically and show similar patterns?

Many of the RBPs fused to a microtubule-binding domain that we tested (FUS, TDP-43, HUR, G3BP-1, YB-1) are not homogenously distributed on microtubules because of the capacity of RBPs to form higher order assemblies when RBPs are confined on microtubules (See Maucuer et al, J Cell Science, 2018, Fig1 B).

- Fig 3c: legend should indicate what the color coding means

OK.

- Figure 5a: LARP6/YB-1 merged image looks yellow, suggesting overlap, but the quantitation doesn't reflect that. Why does it look like YB-1 is in these stress granules? Showing the red and green channels separately below the merged images would be clearer than showing red/green merged and mRNA channels separately. Then the reader could see if YB-1 is concentrated in the red channel in the same place as the stress granule.

LARP-6 and YB-1 indeed co-localize in stress granules. We now show the green and red channel separately. However, the enrichment of YB-1 in stress granules is reduced when LARP-6 is overexpressed compared to Lin28 (quantification).

REVIEWERS' COMMENTS:

Reviewer #1 (Remarks to the Author):

In this revised manuscript, Samsonova have focused their study on the biochemical relationship between the cold-shock domains of YB-1 and Lin28. Although the new manuscript is improved compared to the original submission, many of the fundamental issues remain—overstated conclusions, unclear writing and confusing figures, an over-reliance on coincidence as evidence of their favorite model,—and so I cannot accept publication of this manuscript in its current form. Below I detail some of the issues I find here—but this list is not an extensive one, and I strongly urge the authors to tone down their conclusions throughout the manuscript.

- (1) In much of the manuscript, the authors describe the interaction between YB-1—Lin28—RNA as cooperative. This word has a biochemical meaning; however, no biochemical evidence is presented to demonstrate cooperativity or that it depends upon the CSD (or specific residues identified).
- (2) Similarly, the authors look at the impact of some mutants in co-immunoprecipitation assays and infer “cooperativity.” However, these conclusions are overstated, and more rigorous biochemical experiments need to be done.
- (3) The EMSAs presented are not convincing, especially because the molar ratios are not interpretable without concentrations in the context of Kds.
- (4) The authors write, “In HEK293 cells expressing Lin28-GFP, the presence of endogenous YB-1 was detected in Lin28-GFP-immunoprecipitates without but not with RNase treatment, in agreement with a cooperative association of Lin28 and YB-1 in the presence of mRNA.” An alternative interpretation is Lin28 and YB-1 bind the same transcripts but don’t interact on those mRNAs in vivo. The stated conclusion is a key one for the paper, but it lacks experimental support.
- (5) The authors write, “In contrast with Lin28-GFP, stress granules in cells expressing LARP6 or G3BP1 appeared with a greenish color, which indicates a reduced presence of YB-1 in these stress granules (Fig. 6b). Measurements of the relative enrichment of YB-1 further suggests a better association of YB-1-HA with Lin28 in stress granules compared with G3BP1 and LARP6.” Again, this conclusion is a substantial overinterpretation of the experiments (especially given that the obvious co-localization of YB-1 with G3BP1 and LARP6). There are several other interpretations, and the authors’ conclusion requires additional experimentation.
- (6) In general, the figures and text remain very hard to interpret and are not intuitive.
- (7) Many figures still use a red/green color-scheme and are not accessible.

Reviewer #2 (Remarks to the Author):

Samsonova et al. have provided strong revisions to their manuscript. Specifically, they have:

- (1) added important controls
- (2) they have strengthened their results with additional gel shifts and immunoprecipitation
- (3) they have altered the language in the manuscript to more carefully differentiate between conclusions that can be drawn from their data and new models/hypotheses for the field to consider.

While the paper does not have strong mechanistic insight into the relationship between Lin28 and YB-1, I agree with the authors decision to remove the speculative data on cellular function and leave

that work for another paper. The additional work would have been substantial. However, the interesting trends in co-mixing in vivo, along with the structural studies to identify key residues that affect co-mixing are a nice addition to the field. These results provide a new interesting hypothesis and valuable tools and mutants that can be used to explore the functional consequences of Lin28 and YB-1 interactions on mRNA.

I think the manuscript is ready for publication.

Response to Reviewer's

In this revised manuscript, Samsonova have focused their study on the biochemical relationship between the cold-shock domains of YB-1 and Lin28. Although the new manuscript is improved compared to the original submission, many of the fundamental issues remain—overstated conclusions, unclear writing and confusing figures, an over-reliance on coincidence as evidence of their favorite model,—and so I cannot accept publication of this manuscript in its current form. Below I detail some of the issues I find here—but this list is not an extensive one, and I strongly urge the authors to tone down their conclusions throughout the manuscript.

We have toned down our conclusions throughout the manuscript.

(1) In much of the manuscript, the authors describe the interaction between YB-1—Lin28—RNA as cooperative. This word has a biochemical meaning; however, no biochemical evidence is presented to demonstrate cooperativity or that it depends upon the CSD (or specific residues identified).

(2) Similarly, the authors look at the impact of some mutants in co-immunoprecipitation assays and infer “cooperativity.” However, these conclusions are overstated, and more rigorous biochemical experiments need to be done.

(3) The EMSAs presented are not convincing, especially because the molar ratios are not interpretable without concentrations in the context of Kds.

Remarks (1) (2) (3) are linked.

The gel shift assays presented in Figure 6a and S3a indicate that TDP-43 and FUS are not mixing with YB-1:mRNA complexes (two bands) while a single band is observed in the presence of YB-1 and Lin28. Even if it is not sufficient to ascertain a cooperative association, this result is in agreement with the cellular experiments shown Figure 1a and c with full length proteins which show the mixing between YB-1 and Lin28 in cells. Based on NMR data and MD, we also provide the structural basis (figure 5) of a mechanism that could explain the mixing of YB-1 and Lin28 in the presence of mRNA. The mixing could be explained by a cooperative association of CSD domains along mRNA.

Besides our experimental and MD results, a cooperative binding of the CSD domain to mRNA is not new and has been already described in previous reports both in bacteria and in eukaryotes (Jiang et al, JBC, 1997; Kretov et al., NAR, 2015). Notably, the cooperative association of YB-1 through its cold-shock domain was investigated in details by our group (Kretov et al. NAR, 2015, Figure 3 in this article).

4) The authors write, “In HEK293 cells expressing Lin28-GFP, the presence of endogenous YB-1 was detected in Lin28-GFP-immunoprecipitates without but not with RNase treatment, in agreement with a cooperative association of Lin28 and YB-1 in the presence of mRNA.” An alternative interpretation is Lin28 and YB-1 bind the same transcripts but don't interact on those mRNAs *in vivo*. The stated conclusion is a key one for the paper, but it lacks experimental support.

This is a right remark. Immunoprecipitation cannot tell by itself whether a cooperative binding is taking place or not. This is just an experiment that may rule out a cooperative association, even if we have to be careful about possible protein redistribution during the pull down *in vitro*. However, in addition to IP, we have used NMR data to identify Lin28 residues that are possibly involved in the cooperative

association between Lin28 and YB-1. These residues experienced chemical shift variations and peak broadenings when Lin28 is mixed with YB-1 but not with TDP-43 (Figure 2d). Mutations in these residues (that do not directly participate to the binding of Lin28 to RNA) induced the demixing of Lin28 and YB-1 in cells (Figure 4a,b,c) and a lower efficiency of co-precipitation from cell extracts. In our view, these facts support a cooperative association of Lin28 and YB-1 in mRNPs.

(5) The authors write, “In contrast with Lin28-GFP, stress granules in cells expressing LARP6 or G3BP1 appeared with a greenish color, which indicates a reduced presence of YB-1 in these stress granules (Fig. 6b). Measurements of the relative enrichment of YB-1 further suggests a better association of YB-1-HA with Lin28 in stress granules compared with G3BP1 and LARP6.” Again, this conclusion is a substantial overinterpretation of the experiments (especially given that the obvious co-localization of YB-1 with G3BP1 and LARP6). There are several other interpretations, and the authors’ conclusion requires additional experimentation.

We provided a quantitative measurement of the relative YB-1 enrichment in stress granules that is in agreement with a co-association of Lin28 and YB-1 in stress granules. We do indicate that that other RBPs also co-localize with YB-1 in stress granules. To the mixing between YB-1 and Lin28 in cells can also be clearly observed with the microtubule bench assay that provided a clear-cut result about this point (Figure 1C).

(6) In general, the figures and text remain very hard to interpret and are not intuitive.
(7) Many figures still use a red/green color-scheme and are not accessible.

We tried to improve the text and figures. For all the figures that still use green and red colors, separate schemes are shown for the green and red channels to make them accessible to persons with Red–green color blindness.

Reviewer #2 (Remarks to the Author):

Samsonova et al. have provided strong revisions to their manuscript. Specifically, they have:

(1) added important controls (2) they have strengthened their results with additional gel shifts and immunoprecipitation (3) they have altered the language in the manuscript to more carefully differentiate between conclusions that can be drawn from their data and new models/hypotheses for the field to consider.

While the paper does not have strong mechanistic insight into the relationship between Lin28 and YB-1, I agree with the authors decision to remove the speculative data on cellular function and leave that work for another paper. The additional work would have been substantial. However, the interesting trends in co-mixing in vivo, along with the structural studies to identify key residues that affect co-mixing are a nice addition to the field. These results provide a new interesting hypothesis and valuable tools and mutants that can be used to explore the functional consequences of Lin28 and YB-1 interactions on mRNA.

I think the manuscript is ready for publication.